# Gradient Episodic Memory for Continual Learning

**David Lopez-Paz and Marc'Aurelio Ranzato**
Facebook Artificial Intelligence Research
{dlp,ranzato}@fb.com

## Abstract

One major obstacle towards AI is the poor ability of models to solve new problems quicker, and without forgetting previously acquired knowledge. To better understand this issue, we study the problem of *continual learning*, where the model observes, once and one by one, examples concerning a sequence of tasks. First, we propose a set of metrics to evaluate models learning over a continuum of data. These metrics characterize models not only by their test accuracy, but also in terms of their ability to transfer knowledge across tasks. Second, we propose a model for continual learning, called Gradient Episodic Memory (GEM) that alleviates forgetting, while allowing beneficial transfer of knowledge to previous tasks. Our experiments on variants of the MNIST and CIFAR-100 datasets demonstrate the strong performance of GEM when compared to the state-of-the-art.

## 1 Introduction

The starting point in supervised learning is to collect a *training set* $D_{\text{tr}} = \{(x_i, y_i)\}_{i=1}^n$, where each *example* $(x_i, y_i)$ is composed by a *feature vector* $x_i \in \mathcal{X}$, and a *target vector* $y_i \in \mathcal{Y}$. Most supervised learning methods assume that each example $(x_i, y_i)$ is an identically and independently distributed (iid) sample from a *fixed* probability distribution $P$, which describes a single learning task. The goal of supervised learning is to construct a model $f : \mathcal{X} \to \mathcal{Y}$, used to predict the target vectors $y$ associated to unseen feature vectors $x$, where $(x, y) \sim P$. To accomplish this, supervised learning methods often employ the Empirical Risk Minimization (ERM) principle [Vapnik, 1998], where $f$ is found by minimizing $\frac{1}{|D_{\text{tr}}|} \sum_{(x_i, y_i) \in D_{\text{tr}}} \ell(f(x_i), y_i)$, where $\ell : \mathcal{Y} \times \mathcal{Y} \to [0, \infty)$ is a loss function penalizing prediction errors. In practice, ERM often requires multiple passes over the training set.

ERM is a major simplification from what we deem as human learning. In stark contrast to learning machines, learning humans observe data as an ordered sequence, seldom observe the same example twice, they can only memorize a few pieces of data, and the sequence of examples concerns different learning tasks. Therefore, the iid assumption, along with any hope of employing the ERM principle, fall apart. In fact, straightforward applications of ERM lead to "catastrophic forgetting" [McCloskey and Cohen, 1989]. That is, the learner forgets how to solve past tasks after it is exposed to new tasks.

This paper narrows the gap between ERM and the more human-like learning description above. In particular, our learning machine will observe, *example by example*, the *continuum of data*

$$(x_1, t_1, y_1), \ldots, (x_i, t_i, y_i), \ldots, (x_n, t_n, y_n), \tag{1}$$

where besides input and target vectors, the learner observes $t_i \in \mathcal{T}$, a *task descriptor* identifying the task associated to the pair $(x_i, y_i) \sim P_{t_i}$. Importantly, examples are not drawn iid from a fixed probability distribution over triplets $(x, t, y)$, since a whole sequence of examples from the current task may be observed before switching to the next task. The goal of *continual learning* is to construct a model $f : \mathcal{X} \times \mathcal{T}$ able to predict the target $y$ associated to a test pair $(x, t)$, where $(x, y) \sim P_t$. In this setting, we face challenges unknown to ERM:

1. *Non-iid input data*: the continuum of data is not *iid* with respect to any fixed probability distribution $P(X, T, Y)$ since, once tasks switch, a whole sequence of examples from the new task may be observed.

2. *Catastrophic forgetting*: learning new tasks may hurt the performance of the learner at previously solved tasks.

3. *Transfer learning*: when the tasks in the continuum are related, there exists an opportunity for transfer learning. This would translate into faster learning of new tasks, as well as performance improvements in old tasks.

The rest of this paper is organized as follows. In Section 2, we formalize the problem of continual learning, and introduce a set of metrics to evaluate learners in this scenario. In Section 3, we propose GEM, a model to learn over continuums of data that alleviates forgetting, while transferring beneficial knowledge to past tasks. In Section 4, we compare the performance of GEM to the state-of-the-art. Finally, we conclude by reviewing the related literature in Section 5, and offer some directions for future research in Section 6. Our source code is available at `https://github.com/facebookresearch/GradientEpisodicMemory`.

## 2 A Framework for Continual Learning

We focus on the *continuum* of data of (1), where each triplet $(x_i, t_i, y_i)$ is formed by a feature vector $x_i \in \mathcal{X}_{t_i}$, a task descriptor $t_i \in \mathcal{T}$, and a target vector $y_i \in \mathcal{Y}_{t_i}$. For simplicity, we assume that the continuum is *locally iid*, that is, every triplet $(x_i, t_i, y_i)$ satisfies $(x_i, y_i) \overset{iid}{\sim} P_{t_i}(X, Y)$.

While observing the data (1) *example by example*, our goal is to learn a predictor $f : \mathcal{X} \times \mathcal{T} \to \mathcal{Y}$, which can be queried *at any time* to predict the target vector $y$ associated to a test pair $(x, t)$, where $(x, y) \sim P_t$. Such test pair can belong to a task that we have observed in the past, the current task, or a task that we will experience (or not) in the future.

**Task descriptors** An important component in our framework is the collection of task descriptors $t_1, \ldots, t_n \in \mathcal{T}$. In the simplest case, the task descriptors are integers $t_i = i \in \mathbb{Z}$ enumerating the different tasks appearing in the continuum of data. More generally, task descriptors $t_i$ could be structured objects, such as a paragraph of natural language explaining how to solve the $i$-th task. Rich task descriptors offer an opportunity for zero-shot learning, since the relation between tasks could be inferred using new task descriptors alone. Furthermore, task descriptors disambiguate similar learning tasks. In particular, the same input $x_i$ could appear in two different tasks, but require different targets. Task descriptors can reference the existence of multiple *learning environments*, or provide additional (possibly hierarchical) *contextual information* about each of the examples. However, in this paper we focus on alleviating catastrophic forgetting when learning from a continuum of data, and leave zero-shot learning for future research.

Next, we discuss the training protocol and evaluation metrics for continual learning.

**Training Protocol and Evaluation Metrics**

Most of the literature about learning over a sequence of tasks [Rusu et al., 2016, Fernando et al., 2017, Kirkpatrick et al., 2017, Rebuffi et al., 2017] describes a setting where i) the number of tasks is small, ii) the number of examples per task is large, iii) the learner performs several passes over the examples concerning each task, and iv) the only metric reported is the average performance across all tasks. In contrast, we are interested in the "more human-like" setting where i) the number of tasks is large, ii) the number of training examples per task is small, iii) the learner observes the examples concerning each task only once, and iv) we report metrics that measure both transfer and forgetting.

Therefore, at training time we provide the learner with only one example at the time (or a small mini-batch), in the form of a triplet $(x_i, t_i, y_i)$. The learner never experiences the same example twice, and tasks are streamed in sequence. We do not need to impose any order on the tasks, since a future task may coincide with a past task.

Besides monitoring its performance across tasks, it is also important to assess the ability of the learner to *transfer* knowledge. More specifically, we would like to measure:

1. *Backward transfer* (BWT), which is the influence that learning a task $t$ has on the perfor-
mance on a previous task $k \prec t$. On the one hand, there exists *positive* backward transfer
when learning about some task $t$ increases the performance on some preceding task $k$. On
the other hand, there exists *negative* backward transfer when learning about some task $t$
decreases the performance on some preceding task $k$. Large negative backward transfer is
also known as *(catastrophic) forgetting*.

2. *Forward transfer* (FWT), which is the influence that learning a task $t$ has on the performance
on a future task $k \succ t$. In particular, *positive* forward transfer is possible when the model is
able to perform "zero-shot" learning, perhaps by exploiting the structure available in the
task descriptors.

For a principled evaluation, we consider access to a test set for each of the $T$ tasks. After the model
finishes learning about the task $t_i$, we evaluate its *test* performance on all $T$ tasks. By doing so, we
construct the matrix $R \in \mathbb{R}^{T \times T}$, where $R_{i,j}$ is the test classification accuracy of the model on task $t_j$
after observing the last sample from task $t_i$. Letting $\bar{b}$ be the vector of test accuracies for each task at
random initialization, we define three metrics:

$$\text{Average Accuracy: } \text{ACC} = \frac{1}{T} \sum_{i=1}^{T} R_{T,i} \tag{2}$$

$$\text{Backward Transfer: } \text{BWT} = \frac{1}{T-1} \sum_{i=1}^{T-1} R_{T,i} - R_{i,i} \tag{3}$$

$$\text{Forward Transfer: } \text{FWT} = \frac{1}{T-1} \sum_{i=2}^{T} R_{i-1,i} - \bar{b}_i. \tag{4}$$

The larger these metrics, the better the model. If two models have similar ACC, the most preferable
one is the one with larger BWT and FWT. Note that it is meaningless to discuss backward transfer
for the first task, or forward transfer for the last task.

For a fine-grained evaluation that accounts for learning speed, one can build a matrix $R$ with more
rows than tasks, by evaluating more often. In the extreme case, the number of rows could equal the
number of continuum samples $n$. Then, the number $R_{i,j}$ is the test accuracy on task $t_j$ after observing
the $i$-th example in the continuum. Plotting each column of $R$ results into a learning curve.

## 3 Gradient of Episodic Memory (GEM)

In this section, we propose Gradient Episodic Memory (GEM), a model for continual learning, as
introduced in Section 2. The main feature of GEM is an *episodic memory* $\mathcal{M}_t$, which stores a
subset of the observed examples from task $t$. For simplicity, we assume integer task descriptors, and
use them to index the episodic memory. When using integer task descriptors, one cannot expect
significant positive forward transfer (zero-shot learning). Instead, we focus on minimizing negative
backward transfer (catastrophic forgetting) by the efficient use of episodic memory.

In practice, the learner has a total budget of $M$ memory locations. If the number of total tasks $T$
is known, we can allocate $m = M/T$ memories for each task. Conversely, if the number of total
tasks $T$ is unknown, we can gradually reduce the value of $m$ as we observe new tasks [Rebuffi et al.,
2017]. For simplicity, we assume that the memory is populated with the last $m$ examples from each
task, although better memory update strategies could be employed (such as building a coreset per
task). In the following, we consider predictors $f_\theta$ parameterized by $\theta \in \mathbb{R}^p$, and define the loss at the
memories from the $k$-th task as

$$\ell(f_\theta, \mathcal{M}_k) = \frac{1}{|\mathcal{M}_k|} \sum_{(x_i, k, y_i) \in \mathcal{M}_k} \ell(f_\theta(x_i, k), y_i). \tag{5}$$

Obviously, minimizing the loss at the current example together with (5) results in overfitting to the
examples stored in $\mathcal{M}_k$. As an alternative, we could keep the predictions at past tasks invariant by
means of distillation [Rebuffi et al., 2017]. However, this would deem positive backward transfer
impossible. Instead, we will use the losses (5) as *inequality constraints*, avoiding their increase but

allowing their decrease. In contrast to the state-of-the-art [Kirkpatrick et al., 2017, Rebuffi et al., 2017], our model therefore allows *positive* backward transfer.

More specifically, when observing the triplet $(x, t, y)$, we solve the following problem:

$$\text{minimize}_\theta \quad \ell(f_\theta(x, t), y)$$
$$\text{subject to} \quad \ell(f_\theta, \mathcal{M}_k) \leq \ell(f_\theta^{t-1}, \mathcal{M}_k) \text{ for all } k < t, \tag{6}$$

where $f_\theta^{t-1}$ is the predictor state at the end of learning of task $t - 1$.

In the following, we make two key observations to solve (6) efficiently. First, it is unnecessary to store old predictors $f_\theta^{t-1}$, as long as we guarantee that the loss at previous tasks does not increase after each parameter update $g$. Second, assuming that the function is locally linear (as it happens around small optimization steps) and that the memory is representative of the examples from past tasks, we can diagnose increases in the loss of previous tasks by computing the angle between their loss gradient vector and the proposed update. Mathematically, we rephrase the constraints (6) as:

$$\langle g, g_k \rangle := \left\langle \frac{\partial \ell(f_\theta(x, t), y)}{\partial \theta}, \frac{\partial \ell(f_\theta, \mathcal{M}_k)}{\partial \theta} \right\rangle \geq 0, \text{ for all } k < t. \tag{7}$$

If all the inequality constraints (7) are satisfied, then the proposed parameter update $g$ is unlikely to increase the loss at previous tasks. On the other hand, if one or more of the inequality constraints (7) are violated, then there is at least one previous task that would experience an increase in loss after the parameter update. If violations occur, we propose to project the proposed gradient $g$ to the closest gradient $\tilde{g}$ (in squared $\ell_2$ norm) satisfying all the constraints (7). Therefore, we are interested in:

$$\text{minimize}_{\tilde{g}} \frac{1}{2} \quad \|g - \tilde{g}\|_2^2$$
$$\text{subject to} \quad \langle \tilde{g}, g_k \rangle \geq 0 \text{ for all } k < t. \tag{8}$$

To solve (8) efficiently, recall the primal of a Quadratic Program (QP) with inequality constraints:

$$\text{minimize}_z \quad \frac{1}{2} z^\top C z + p^\top z$$
$$\text{subject to} \quad Az \geq b, \tag{9}$$

where $C \in \mathbb{R}^{p \times p}$, $p \in \mathbb{R}^p$, $A \in \mathbb{R}^{(t-1) \times p}$, and $b \in \mathbb{R}^{t-1}$. The dual problem of (9) is:

$$\text{minimize}_{u,v} \quad \frac{1}{2} u^\top C u - b^\top v$$
$$\text{subject to} \quad A^\top v - C u = p,$$
$$v \geq 0. \tag{10}$$

If $(u^\star, v^\star)$ is a solution to (10), then there is a solution $z^\star$ to (9) satisfying $Cz^\star = Cu^\star$ [Dorn, 1960]. Quadratic programs are at the heart of support vector machines [Scholkopf and Smola, 2001].

With these notations in hand, we write the primal GEM QP (8) as:

$$\text{minimize}_z \quad \frac{1}{2} z^\top z - g^\top z + \frac{1}{2} g^\top g$$
$$\text{subject to} \quad Gz \geq 0,$$

where $G = -(g_1, \ldots, g_{t-1})$, and we discard the constant term $g^\top g$. This is a QP on $p$ variables (the number of parameters of the neural network), which could be measured in the millions. However, we can pose the dual of the GEM QP as:

$$\text{minimize}_v \quad \frac{1}{2} v^\top G G^\top v + g^\top G^\top v$$
$$\text{subject to} \quad v \geq 0, \tag{11}$$

since $u = G^\top v + g$ and the term $g^\top g$ is constant. This is a QP on $t - 1 \ll p$ variables, the number of observed tasks so far. Once we solve the dual problem (11) for $v^\star$, we can recover the projected gradient update as $\tilde{g} = G^\top v^\star + g$. In practice, we found that adding a small constant $\gamma \geq 0$ to $v^\star$ biased the gradient projection to updates that favoured benefitial backwards transfer.

Algorithm 1 summarizes the training and evaluation protocol of GEM over a continuum of data. The pseudo-code includes the computation of the matrix R, containing the sufficient statistics to compute the metrics ACC, FWT, and BWT described in Section 2.

**A causal compression view**   We can interpret GEM as a model that learns the subset of correlations common to a set of distributions (tasks). Furthermore, GEM can (and will in our MNIST experiments) be used to predict target vectors associated to previous or new tasks without making use of task descriptors. This is a desired feature in causal inference problems, since causal predictions are invariant across different environments [Peters et al., 2016], and therefore provide the most compressed representation of a set of distributions [Schölkopf et al., 2016].

---

**Algorithm 1** Training a GEM over an *ordered* continuum of data
___

**procedure** TRAIN($f_\theta$, Continuum$_{\text{train}}$, Continuum$_{\text{test}}$)
    $\mathcal{M}_t \leftarrow \{\}$ for all $t = 1, \ldots, T$.
    $R \leftarrow 0 \in \mathbb{R}^{T \times T}$.
    **for** $t = 1, \ldots, T$ **do**:
        **for** $(x, y)$ in Continuum$_{\text{train}}(t)$ **do**
            $\mathcal{M}_t \leftarrow \mathcal{M}_t \cup (x, y)$
            $g \leftarrow \nabla_\theta \, \ell(f_\theta(x, t), y)$
            $g_k \leftarrow \nabla_\theta \, \ell(f_\theta, \mathcal{M}_k)$ for all $k < t$
            $\tilde{g} \leftarrow$ PROJECT($g, g_1, \ldots, g_{t-1}$), see (11).
            $\theta \leftarrow \theta - \alpha \tilde{g}$.
        **end for**
        $R_{t,:} \leftarrow$ EVALUATE($f_\theta$, Continuum$_{\text{test}}$)
    **end for**
    **return** $f_\theta$, R
**end procedure**

**procedure** EVALUATE($f_\theta$, Continuum)
    $r \leftarrow 0 \in \mathbb{R}^T$
    **for** $k = 1, \ldots, T$ **do**
        $r_k \leftarrow 0$
        **for** $(x, y)$ in Continuum($k$) **do**
            $r_k \leftarrow r_k +$ accuracy($f_\theta(x, k), y$)
        **end for**
        $r_k \leftarrow r_k \,/$ len(Continuum($k$))
    **end for**
    **return** $r$
**end procedure**

---

# 4   Experiments

We perform a variety of experiments to assess the performance of GEM in continual learning.

## 4.1   Datasets

We consider the following datasets:

- *MNIST Permutations* [Kirkpatrick et al., 2017], a variant of the MNIST dataset of handwritten digits [LeCun et al., 1998], where each task is transformed by a fixed permutation of pixels. In this dataset, the input distribution for each task is unrelated.

- *MNIST Rotations*, a variant of MNIST where each task contains digits rotated by a fixed angle between 0 and 180 degrees.

- *Incremental CIFAR100* [Rebuffi et al., 2017], a variant of the CIFAR object recognition dataset with 100 classes [Krizhevsky, 2009], where each task introduces a new set of classes. For a total number of $T$ tasks, each new task concerns examples from a disjoint subset of $100/T$ classes. Here, the input distribution is similar for all tasks, but different tasks require different output distributions.

For all the datasets, we considered $T = 20$ tasks. On the MNIST datasets, each task has 1000 examples from 10 different classes. On the CIFAR100 dataset each task has 2500 examples from 5 different classes. The model observes the tasks in sequence, and each example once. The evaluation for each task is performed on the test partition of each dataset.

## 4.2   Architectures

On the MNIST tasks, we use fully-connected neural networks with two hidden layers of 100 ReLU units. On the CIFAR100 tasks, we use a smaller version of ResNet18 [He et al., 2015], with three times less feature maps across all layers. Also on CIFAR100, the network has a final linear classifier per task. This is one simple way to leverage the task descriptor, in order to adapt the output distribution to the subset of classes for each task. We train all the networks and baselines using plain SGD on mini-batches of 10 samples. All hyper-parameters are optimized using a grid-search (see Appendix A), and the best results for each model are reported.

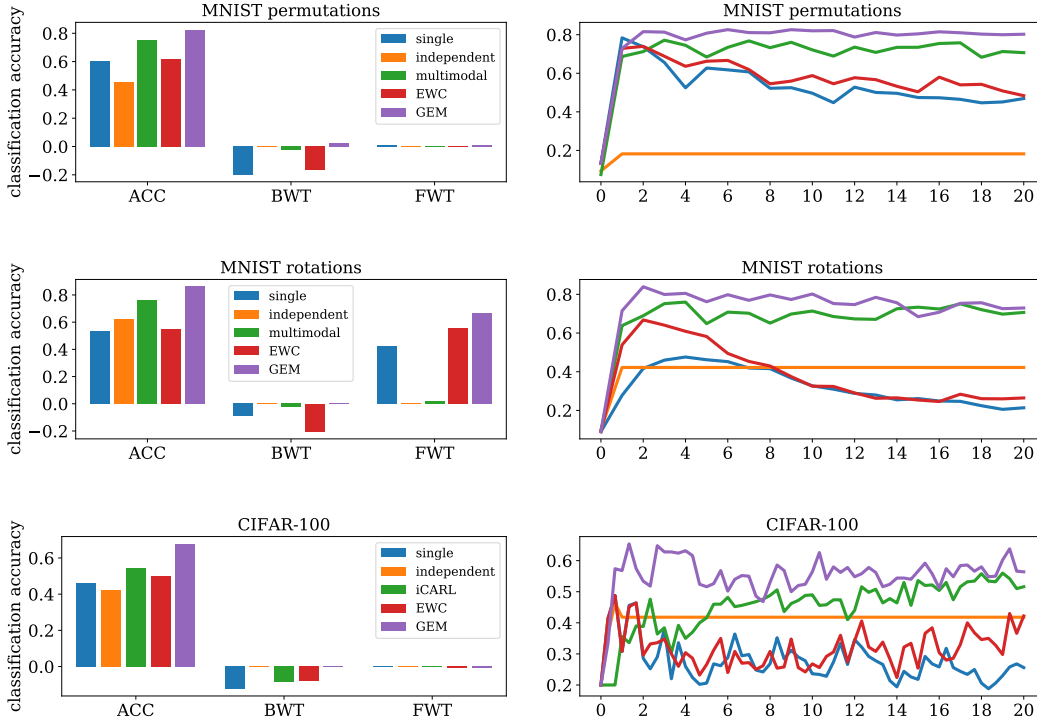

Figure 1: Left: ACC, BWT, and FWT for all datasets and methods. Right: evolution of the test accuracy at the first task, as more tasks are learned.

Table 1: CPU Training time (s) of MNIST experiments for all methods.

| task | single | independent | multimodal | EWC | GEM |
|---|---|---|---|---|---|
| permutations | 11 | 11 | 14 | 179 | 77 |
| rotations | 11 | 16 | 13 | 169 | 135 |

## 4.3 Methods

We compare GEM to five alternatives:

1. a *single* predictor trained across all tasks.

2. one *independent* predictor per task. Each *independent* predictor has the same architecture as "single" but with $T$ times less hidden units than "single". Each new independent predictor can be initialized at random, or be a clone of the last trained predictor (decided by grid-search).

3. a *multimodal* predictor, which has the same architecture of "single", but with a dedicated input layer per task (only for MNIST datasets).

4. *EWC* [Kirkpatrick et al., 2017], where the loss is regularized to avoid catastrophic forgetting.

5. *iCARL* [Rebuffi et al., 2017], a class-incremental learner that classifies using a nearest-exemplar algorithm, and prevents catastrophic forgetting by using an episodic memory. iCARL requires the same input representation across tasks, so this method only applies to our experiment on CIFAR100.

GEM, iCaRL and EWC have the same architecture as "single", plus episodic memory.

Table 2: ACC as a function of the episodic memory size for GEM and iCARL, on CIFAR100.

| memory size | 200 | 1,280 | 2,560 | 5,120 |
|---|---|---|---|---|
| GEM | 0.487 | 0.579 | 0.633 | 0.654 |
| iCARL | 0.436 | 0.494 | 0.500 | 0.508 |

Table 3: ACC/BWT on the MNIST Rotations dataset, when varying the number of epochs per task.

| method | 1 epoch | 2 epochs | 5 epochs |
|---|---|---|---|
| single, shuffled data | 0.83/ | 0.87/ | 0.89/ |
| single | 0.53/-0.08 | 0.49/-0.25 | 0.43/-0.40 |
| independent | 0.56/-0.00 | 0.64/-0.00 | 0.67/-0.00 |
| multimodal | 0.76/-0.02 | 0.72/-0.11 | 0.59/-0.28 |
| EWC | 0.55/-0.19 | 0.59/-0.17 | 0.61/-0.11 |
| GEM | 0.86/+0.05 | 0.88/+0.02 | 0.89/-0.02 |

## 4.4 Results

Figure 1 (left) summarizes the average accuracy (ACC, Equation 2), backward transfer (BWT, Equation 3) and forward transfer (FWT, Equation 4) for all datasets and methods. We provide the full evaluation matrices $R$ in Appendix B. Overall, GEM performs similarly or better than the multimodal model (which is very well suited to the MNIST tasks). GEM minimizes backward transfer, while exhibiting negligible or positive forward transfer.

Figure 1 (right) shows the evolution of the test accuracy of the first task throughout the continuum of data. GEM exhibits minimal forgetting, and positive backward transfer in CIFAR100.

Overall, GEM performs significantly better than other continual learning methods like EWC, while spending less computation (Table 1). GEM's efficiency comes from optimizing over a number of variables equal to the number of tasks ($T = 20$ in our experiments), instead of optimizing over a number of variables equal to the number of parameters ($p = 1109240$ for CIFAR100 for instance). GEM's bottleneck is the necessity of computing previous task gradients at each learning iteration.

### 4.4.1 Importance of memory, number of passes, and order of tasks

Table 2 shows the final ACC in the CIFAR-100 experiment for both GEM and iCARL as a function their episodic memory size. Also seen in Table 2, the final ACC of GEM is an increasing function of the size of the episodic memory, eliminating the need to carefully tune this hyper-parameter. GEM outperforms iCARL for a wide range of memory sizes.

Table 3 illustrates the importance of memory as we do more than one pass through the data on the MNIST rotations experiment. Multiple training passe exacerbate the catastrophic forgetting problem. For instance, in the last column of Table 3 (except for the result in the first row), each model is shown examples of a task five times (in random order) before switching to the next task. Table 3 shows that memory-less methods (like "single" and "multimodal") exhibit higher negative BWT, leading to lower ACC. On the other hand, memory-based methods such as EWC and GEM lead to higher ACC as the number of passes through the data increases. However, GEM suffers less negative BWT than EWC, leading to a higher ACC.

Finally, to relate the performance of GEM to the best possible performance on the proposed datasets, the first row of Table 3 reports the ACC of "single" when trained with iid data from all tasks. This mimics usual multi-task learning, where each mini-batch contains examples taken from a random selection of tasks. By comparing the first and last row of Table 3, we see that GEM matches the "oracle performance upper-bound" ACC provided by iid learning, and minimizes negative BWT.

# 5 Related work

Continual learning [Ring, 1994], also called *lifelong learning* [Thrun, 1994, Thrun and Pratt, 2012, Thrun, 1998, 1996], considers learning through a sequence of tasks, where the learner has to retain knowledge about past tasks and leverage that knowledge to quickly acquire new skills. This learning setting led to implementations [Carlson et al., 2010, Ruvolo and Eaton, 2013, Ring, 1997], and theoretical investigations [Baxter, 2000, Balcan et al., 2015, Pentina and Urner, 2016], although the latter ones have been restricted to linear models. In this work, we revisited continual learning but proposed to focus on the more realistic setting where examples are seen only once, memory is finite, and the learner is also provided with (potentially structured) task descriptors. Within this framework, we introduced a new set of metrics, a training and testing protocol, and a new algorithm, GEM, that outperforms the current state-of-the-art in terms of limiting forgetting.

The use of task descriptors is similar in spirit to recent work in Reinforcement Learning [Sutton et al., 2011, Schaul et al., 2015], where task or goal descriptors are also fed as input to the system. The *CommAI project* [Mikolov et al., 2015, Baroni et al., 2017] shares our same motivations, but focuses on highly structured task descriptors, such as strings of text. In contrast, we focus on the problem of catastrophic forgetting [McCloskey and Cohen, 1989, French, 1999, Ratcliff, 1990, McClelland et al., 1995, Goodfellow et al., 2013].

Several approaches have been proposed to avoid catastrophic forgetting. The simplest approach in neural networks is to freeze early layers, while cloning and fine-tuning later layers on the new task [Oquab et al., 2014] (which we considered in our "independent" baseline). This relates to methods that leverage a modular structure of the network with primitives that can be shared across tasks [Rusu et al., 2016, Fernando et al., 2017, Aljundi et al., 2016, Denoyer and Gallinari, 2015, Eigen et al., 2014]. Unfortunately, it has been very hard to scale up these methods to lots of modules and tasks, given the combinatorial number of compositions of modules.

Our approach is most similar to the regularization approaches that consider a single model, but modify its learning objective to prevent catastrophic forgetting. Within this class of methods, there are approaches that leverage "synaptic" memory [Kirkpatrick et al., 2017, Zenke et al., 2017], where learning rates are adjusted to minimize changes in parameters important for previous tasks. Other approaches are instead based on "episodic" memory [Jung et al., 2016, Li and Hoiem, 2016, Rannen Triki et al., 2017, Rebuffi et al., 2017], where examples from previous tasks are stored and replayed to maintain predictions invariant by means of distillation [Hinton et al., 2015]. GEM is related to these latter approaches but, unlike them, allows for positive backward transfer.

More generally, there are a variety of setups in the machine learning literature related to continual learning. *Multitask learning* [Caruana, 1998] considers the problem of maximizing the performance of a learning machine across a variety of tasks, but the setup assumes simultaneous access to all the tasks at once. Similarly, *transfer learning* [Pan and Yang, 2010] and *domain adaptation* [Ben-David et al., 2010] assume the simultaneous availability of multiple learning tasks, but focus at improving the performance at one of them in particular. *Zero-shot learning* [Lampert et al., 2009, Palatucci et al., 2009] and *one-shot learning* [Fei-Fei et al., 2003, Vinyals et al., 2016, Santoro et al., 2016, Bertinetto et al., 2016] aim at performing well on unseen tasks, but ignore the catastrophic forgetting of previously learned tasks. *Curriculum learning* considers learning a sequence of data [Bengio et al., 2009], or a sequence of tasks [Pentina et al., 2015], sorted by increasing difficulty.

# 6 Conclusion

We formalized the scenario of *continual learning*. First, we defined training and evaluation protocols to assess the quality of models in terms of their *accuracy*, as well as their ability to transfer knowledge *forward* and *backward* between tasks. Second, we introduced GEM, a simple model that leverages an episodic memory to avoid forgetting and favor positive backward transfer. Our experiments demonstrate the competitive performance of GEM against the state-of-the-art.

GEM has three points for improvement. First, GEM does not leverage structured task descriptors, which may be exploited to obtain positive forward transfer (zero-shot learning). Second, we did not investigate advanced memory management (such as building *coresets of tasks* [Lucic et al., 2017]). Third, each GEM iteration requires one backward pass per task, increasing computation time. These are exciting research directions to extend learning machines beyond ERM, and to continuums of data.

## Acknowledgements

We are grateful to M. Baroni, L. Bottou, M. Nickel, Y. Olivier and A. Szlam for their insight. We are grateful to Martin Arjovsky for the QP interpretation of GEM.

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
