[Supplementary Material · appendix.pdf]

# Supplementary material for:
# *Gradient Episodic Memory for Continual Learning*

**David Lopez-Paz and Marc'Aurelio Ranzato**
Facebook Artificial Intelligence Research
{dlp,ranzato}@fb.com

## A Hyper-parameter Selection

Here we report the hyper-parameter grids considered in our experiments. The best values for the MNIST rotations (rot), MNIST permutations (perm) and CIFAR-100 incremental (cifar) experiments are noted accordingly in parenthesis. For more details, please refer to our implementation, linked in the main text.

- *single*
  - learning rate: [0.001, 0.003 (rot), 0.01, 0.03 (perm), 0.1, 0.3, 1.0 (cifar)]
- *independent*
  - learning rate: [0.001, 0.003, 0.01, 0.03 (perm), 0.1 (rot), 0.3 (cifar), 1.0]
  - finetune: [no, yes (rot, perm, cifar)]
- *multimodal*
  - learning rate: [0.001, 0.003, 0.01, 0.03, 0.1 (rot, perm), 0.3, 1.0]
- *EWC*
  - learning rate: [0.001, 0.003, 0.01 (rot), 0.03, 0.1 (perm), 0.3, 1.0 (cifar)]
  - regularization: [1 (cifar), 3 (perm), 10, 30, 100, 300, 1000 (rot), 3000, 10000, 30000]
- *iCARL*
  - learning rate: [0.001, 0.003, 0.01, 0.03, 0.1, 0.3, 1.0 (cifar)]
  - regularization: [0.1, 0.3, 1 (cifar), 3, 10, 30]
  - memory size: [200, 1280, 2560, 5120 (cifar)]
- *GEM*
  - learning rate: [0.001, 0.003, 0.01, 0.03, 0.1 (rot, perm, cifar), 0.3, 1.0]
  - memory size: [5120 (rot, perm, cifar)]
  - $\gamma$: [0.0, 0.1, ..., 0.5 (rot, perm, cifar), ..., 1.0]

# B  Full experiments

In this section we report the evaluation matrices $R$ for each model and dataset. The first row of each matrix (above the line) is the baseline test accuracy $\bar{b}$ before training starts. The rest of entries $(i, j)$ of the matrix $R$ report the test accuracy of the $j$-th task just after finishing training the $i$-th task.

## B.1  MNIST permutations

### B.1.1  Model *single*

```
0.1330 0.1199 0.1070 0.0825 0.0609 0.0832 0.1385 0.1123 0.0736 0.1190 0.0666 0.0890 0.0885 0.0723 0.1083 0.0524 0.0976 0.0871 0.1143 0.0743
|
0.7838 0.1264 0.0709 0.1142 0.0671 0.0928 0.0925 0.0878 0.0759 0.0832 0.0748 0.1057 0.0927 0.0770 0.0886 0.0995 0.1328 0.0839 0.1802 0.0609
0.7374 0.7939 0.0981 0.1496 0.0859 0.0626 0.1226 0.0666 0.1045 0.0596 0.0787 0.0938 0.0948 0.0737 0.0797 0.0670 0.1450 0.1062 0.1438 0.0714
0.6564 0.7196 0.7783 0.1177 0.0603 0.0908 0.1296 0.0654 0.1246 0.0894 0.0721 0.0798 0.0811 0.0631 0.0759 0.0726 0.1338 0.1265 0.1567 0.1039
0.5246 0.6359 0.7593 0.7790 0.0612 0.0864 0.1118 0.0815 0.1028 0.0900 0.0991 0.0773 0.0947 0.0948 0.0931 0.0930 0.1111 0.1027 0.1504 0.1003
0.6272 0.7265 0.7737 0.7695 0.7827 0.0981 0.0898 0.0530 0.1031 0.0760 0.0675 0.0569 0.0989 0.0698 0.0810 0.0483 0.1305 0.0819 0.1386 0.0961
0.6177 0.7317 0.7890 0.7860 0.7808 0.8135 0.1020 0.0668 0.1005 0.0648 0.0456 0.0705 0.0932 0.0858 0.1056 0.0797 0.1483 0.1039 0.1442 0.0788
0.6071 0.6620 0.7623 0.7540 0.7657 0.7838 0.8113 0.0991 0.1127 0.0957 0.0493 0.0916 0.0713 0.0804 0.1383 0.0639 0.1336 0.1192 0.1336 0.0921
0.5221 0.6675 0.7466 0.6990 0.7239 0.7707 0.8156 0.7909 0.1107 0.0858 0.0510 0.0764 0.0668 0.0849 0.1103 0.0441 0.1232 0.0996 0.1434 0.0744
0.5252 0.6347 0.6880 0.6490 0.7100 0.7478 0.7958 0.7595 0.8149 0.0981 0.0514 0.0843 0.0658 0.0922 0.1139 0.0529 0.1113 0.0846 0.1321 0.0897
0.4967 0.5864 0.6611 0.6384 0.6439 0.7252 0.7579 0.7433 0.7643 0.7761 0.0757 0.0799 0.0757 0.0864 0.0959 0.0685 0.1234 0.0733 0.1842 0.0930
0.4475 0.5803 0.6970 0.6617 0.6656 0.7274 0.7326 0.7567 0.7699 0.7763 0.8150 0.0869 0.0600 0.1025 0.1103 0.0537 0.1281 0.1026 0.2042 0.0885
0.5281 0.5656 0.6930 0.5233 0.6089 0.6503 0.7072 0.7421 0.7283 0.7588 0.7663 0.7865 0.0570 0.1119 0.1056 0.0495 0.1348 0.0943 0.1699 0.0789
0.5007 0.5260 0.6689 0.5819 0.5772 0.5277 0.6928 0.7165 0.7096 0.7229 0.7449 0.7382 0.8271 0.1264 0.1036 0.0706 0.1131 0.0985 0.1906 0.0820
0.4961 0.4526 0.6794 0.5465 0.5234 0.5238 0.6348 0.6833 0.6908 0.6886 0.7163 0.7292 0.7810 0.7952 0.1053 0.0661 0.1135 0.0942 0.2164 0.0806
0.4743 0.3400 0.5585 0.5611 0.5606 0.5228 0.6122 0.5523 0.6552 0.6740 0.6804 0.7231 0.7709 0.7340 0.8112 0.0747 0.1137 0.0811 0.1933 0.0988
0.4730 0.3533 0.5391 0.4235 0.4580 0.4516 0.6392 0.4629 0.6467 0.6725 0.6265 0.7099 0.7861 0.7021 0.7690 0.8066 0.0756 0.0953 0.1826 0.1147
0.4648 0.3295 0.5033 0.3883 0.4151 0.4542 0.5422 0.4495 0.5977 0.6719 0.5681 0.6966 0.7251 0.6391 0.7507 0.7260 0.7925 0.1094 0.1796 0.1389
0.4468 0.3238 0.4941 0.3709 0.4352 0.4632 0.5464 0.4534 0.6094 0.6366 0.6202 0.6787 0.6911 0.6615 0.7107 0.7206 0.8035 0.8109 0.1575 0.1526
0.4513 0.2867 0.4935 0.3826 0.4880 0.4347 0.5075 0.4067 0.5570 0.5492 0.5810 0.6007 0.6724 0.6000 0.7521 0.6801 0.7754 0.7353 0.8174 0.1532
0.4690 0.3482 0.5277 0.3742 0.4973 0.4406 0.5743 0.4568 0.6527 0.5518 0.6356 0.6396 0.6216 0.6537 0.7388 0.7072 0.8005 0.7300 0.8062 0.8107
```

```
Final Accuracy: 0.6018
Backward: -0.1980
Forward:  0.0093
```

### B.1.2  Model *independent*

```
0.0936 0.0995 0.0884 0.0893 0.0784 0.1000 0.1108 0.0965 0.1243 0.1048 0.0819 0.1115 0.0999 0.0762 0.1060 0.1260 0.0930 0.1075 0.1126 0.1092
|
0.1821 0.0995 0.0884 0.0893 0.0784 0.1000 0.1108 0.0965 0.1243 0.1048 0.0819 0.1115 0.0999 0.0762 0.1060 0.1260 0.0930 0.1075 0.1126 0.1092
0.1821 0.2601 0.0884 0.0893 0.0784 0.1000 0.1108 0.0965 0.1243 0.1048 0.0819 0.1115 0.0999 0.0762 0.1060 0.1260 0.0930 0.1075 0.1126 0.1092
0.1821 0.2601 0.3538 0.0893 0.0784 0.1000 0.1108 0.0965 0.1243 0.1048 0.0819 0.1115 0.0999 0.0762 0.1060 0.1260 0.0930 0.1075 0.1126 0.1092
0.1821 0.2601 0.3538 0.3089 0.0784 0.1000 0.1108 0.0965 0.1243 0.1048 0.0819 0.1115 0.0999 0.0762 0.1060 0.1260 0.0930 0.1075 0.1126 0.1092
0.1821 0.2601 0.3538 0.3089 0.4031 0.1000 0.1108 0.0965 0.1243 0.1048 0.0819 0.1115 0.0999 0.0762 0.1060 0.1260 0.0930 0.1075 0.1126 0.1092
0.1821 0.2601 0.3538 0.3089 0.4031 0.3267 0.1108 0.0965 0.1243 0.1048 0.0819 0.1115 0.0999 0.0762 0.1060 0.1260 0.0930 0.1075 0.1126 0.1092
0.1821 0.2601 0.3538 0.3089 0.4031 0.3267 0.4071 0.0965 0.1243 0.1048 0.0819 0.1115 0.0999 0.0762 0.1060 0.1260 0.0930 0.1075 0.1126 0.1092
0.1821 0.2601 0.3538 0.3089 0.4031 0.3267 0.4071 0.4030 0.1243 0.1048 0.0819 0.1115 0.0999 0.0762 0.1060 0.1260 0.0930 0.1075 0.1126 0.1092
0.1821 0.2601 0.3538 0.3089 0.4031 0.3267 0.4071 0.4030 0.4495 0.1048 0.0819 0.1115 0.0999 0.0762 0.1060 0.1260 0.0930 0.1075 0.1126 0.1092
0.1821 0.2601 0.3538 0.3089 0.4031 0.3267 0.4071 0.4030 0.4495 0.5276 0.0819 0.1115 0.0999 0.0762 0.1060 0.1260 0.0930 0.1075 0.1126 0.1092
0.1821 0.2601 0.3538 0.3089 0.4031 0.3267 0.4071 0.4030 0.4495 0.5276 0.4779 0.1115 0.0999 0.0762 0.1060 0.1260 0.0930 0.1075 0.1126 0.1092
0.1821 0.2601 0.3538 0.3089 0.4031 0.3267 0.4071 0.4030 0.4495 0.5276 0.4779 0.5357 0.0999 0.0762 0.1060 0.1260 0.0930 0.1075 0.1126 0.1092
0.1821 0.2601 0.3538 0.3089 0.4031 0.3267 0.4071 0.4030 0.4495 0.5276 0.4779 0.5357 0.5393 0.0762 0.1060 0.1260 0.0930 0.1075 0.1126 0.1092
0.1821 0.2601 0.3538 0.3089 0.4031 0.3267 0.4071 0.4030 0.4495 0.5276 0.4779 0.5357 0.5393 0.5755 0.1060 0.1260 0.0930 0.1075 0.1126 0.1092
0.1821 0.2601 0.3538 0.3089 0.4031 0.3267 0.4071 0.4030 0.4495 0.5276 0.4779 0.5357 0.5393 0.5755 0.5592 0.1260 0.0930 0.1075 0.1126 0.1092
0.1821 0.2601 0.3538 0.3089 0.4031 0.3267 0.4071 0.4030 0.4495 0.5276 0.4779 0.5357 0.5393 0.5755 0.5592 0.5145 0.0930 0.1075 0.1126 0.1092
0.1821 0.2601 0.3538 0.3089 0.4031 0.3267 0.4071 0.4030 0.4495 0.5276 0.4779 0.5357 0.5393 0.5755 0.5592 0.5145 0.5553 0.1075 0.1126 0.1092
0.1821 0.2601 0.3538 0.3089 0.4031 0.3267 0.4071 0.4030 0.4495 0.5276 0.4779 0.5357 0.5393 0.5755 0.5592 0.5145 0.5553 0.5675 0.1126 0.1092
0.1821 0.2601 0.3538 0.3089 0.4031 0.3267 0.4071 0.4030 0.4495 0.5276 0.4779 0.5357 0.5393 0.5755 0.5592 0.5145 0.5553 0.5675 0.5252 0.1092
0.1821 0.2601 0.3538 0.3089 0.4031 0.3267 0.4071 0.4030 0.4495 0.5276 0.4779 0.5357 0.5393 0.5755 0.5592 0.5145 0.5553 0.5675 0.5252 0.5739
```

```
Final Accuracy: 0.4523
Backward: 0.0000
Forward:  0.0000
```

### B.1.3  Model *multimodal*

```
0.0749 0.1152 0.0601 0.0885 0.0826 0.0856 0.0925 0.0703 0.1079 0.0891 0.1029 0.1092 0.0866 0.1014 0.1575 0.1005 0.1083 0.1038 0.0857 0.0759
|
0.6871 0.1306 0.1402 0.1209 0.1108 0.1135 0.1195 0.1066 0.1368 0.1217 0.0605 0.1168 0.1094 0.0970 0.1150 0.1322 0.1471 0.1243 0.1054 0.1567
0.7123 0.8211 0.1344 0.1197 0.0923 0.1155 0.1402 0.1030 0.1346 0.1424 0.1030 0.1329 0.0910 0.0875 0.1437 0.1245 0.1094 0.0865 0.0771 0.1768
0.7716 0.8180 0.8163 0.1208 0.0871 0.1154 0.1308 0.0916 0.1482 0.1238 0.1092 0.1159 0.1122 0.0868 0.1192 0.1221 0.1518 0.0981 0.0692 0.1362
0.7451 0.7961 0.7976 0.7757 0.0883 0.1152 0.1235 0.0849 0.1612 0.1265 0.0976 0.1177 0.1112 0.0914 0.1177 0.1072 0.1681 0.0847 0.0724 0.1495
0.6841 0.7759 0.7569 0.7709 0.7887 0.1147 0.1296 0.0749 0.1561 0.1259 0.0958 0.1216 0.1104 0.0878 0.1261 0.1410 0.1496 0.1009 0.0741 0.1783
0.7344 0.8274 0.8126 0.7832 0.8041 0.8057 0.1145 0.0647 0.1449 0.1446 0.0826 0.1047 0.0862 0.0896 0.1324 0.1362 0.1217 0.0773 0.0878 0.1640
0.7679 0.8373 0.8228 0.7891 0.8117 0.7830 0.8464 0.0673 0.1467 0.1437 0.0831 0.1015 0.0864 0.0880 0.1340 0.1264 0.1212 0.0763 0.0819 0.1532
0.7327 0.8004 0.7749 0.7608 0.7858 0.8231 0.8253 0.7830 0.1472 0.1343 0.1082 0.1209 0.1179 0.1115 0.1159 0.1263 0.1357 0.0862 0.0988 0.1824
0.7606 0.7713 0.7570 0.7962 0.7871 0.8118 0.8393 0.7462 0.7877 0.1288 0.0877 0.1018 0.0819 0.0664 0.1287 0.1060 0.1107 0.0679 0.0801 0.1169
0.7213 0.7791 0.7889 0.7818 0.7401 0.7756 0.8084 0.7793 0.7996 0.7521 0.0575 0.1151 0.0893 0.0667 0.1280 0.1092 0.1194 0.0717 0.0732 0.1365
0.6891 0.7691 0.7889 0.7772 0.7392 0.8103 0.7917 0.7584 0.8111 0.7634 0.8079 0.0984 0.0758 0.0815 0.1398 0.1195 0.1019 0.0849 0.0783 0.1104
0.7357 0.8039 0.7822 0.7710 0.7673 0.8090 0.7918 0.8125 0.7698 0.7689 0.7859 0.7383 0.0858 0.0681 0.1369 0.1135 0.1170 0.0762 0.0780 0.1330
0.7082 0.7856 0.7612 0.7351 0.7639 0.7871 0.7741 0.8013 0.7273 0.7818 0.7458 0.6813 0.8387 0.0607 0.1368 0.1084 0.1106 0.0686 0.0696 0.1195
0.7344 0.7662 0.7261 0.7582 0.7844 0.8025 0.8005 0.7910 0.7397 0.7952 0.7445 0.6660 0.8303 0.6274 0.1447 0.1076 0.1078 0.0711 0.0655 0.1039
```

0.7347 0.7716 0.7370 0.7814 0.7901 0.7909 0.8188 0.7887 0.7521 0.7739 0.7428 0.6771 0.8373 0.6218 0.8413 0.1006 0.0890 0.0667 0.0688 0.0922
0.7544 0.7908 0.7497 0.8055 0.7880 0.8026 0.8243 0.7854 0.7665 0.7741 0.7767 0.7199 0.8211 0.6111 0.8349 0.7976 0.0985 0.0813 0.0811 0.0934
0.7579 0.7631 0.7103 0.7976 0.7695 0.7876 0.8066 0.7507 0.7860 0.7708 0.7680 0.6620 0.7952 0.5357 0.8083 0.7646 0.7769 0.0813 0.0730 0.0990
0.6829 0.7751 0.7282 0.7604 0.7526 0.7427 0.7484 0.7788 0.7369 0.7357 0.7413 0.6109 0.8131 0.5895 0.8224 0.7750 0.8211 0.7871 0.0771 0.0881
0.7128 0.7917 0.7226 0.7779 0.7647 0.7361 0.8046 0.7454 0.7549 0.7559 0.7757 0.6850 0.8092 0.5882 0.8022 0.7528 0.8176 0.7988 0.7844 0.0902
0.7068 0.7832 0.7106 0.7704 0.7642 0.7382 0.7866 0.7218 0.7623 0.7496 0.7829 0.7072 0.8015 0.5605 0.7974 0.7645 0.8115 0.8045 0.7904 0.8077

Final Accuracy: 0.7561
Backward: -0.0275
Forward:  0.0059

### B.1.4   Model *EWC*

0.1330 0.1199 0.1070 0.0825 0.0609 0.0832 0.1385 0.1123 0.0736 0.1190 0.0666 0.0890 0.0885 0.0723 0.1083 0.0524 0.0976 0.0871 0.1143 0.0743
|
0.7289 0.1426 0.0776 0.1272 0.0637 0.0582 0.0944 0.0961 0.0691 0.0794 0.0822 0.1426 0.0953 0.0792 0.0885 0.1094 0.1328 0.0987 0.1507 0.0976
0.7396 0.8305 0.0703 0.1585 0.0749 0.0545 0.1558 0.0789 0.1076 0.0857 0.0699 0.1361 0.0953 0.0581 0.0704 0.0850 0.1241 0.1280 0.1382 0.1012
0.6882 0.7731 0.8274 0.1326 0.0496 0.0917 0.1571 0.0767 0.1015 0.1061 0.0790 0.0712 0.0622 0.0632 0.0982 0.0803 0.1535 0.1088 0.1399 0.1134
0.6361 0.7054 0.7915 0.7780 0.0733 0.0681 0.1446 0.0587 0.0888 0.1046 0.1141 0.0731 0.1008 0.0928 0.0968 0.1183 0.1263 0.1112 0.1130 0.1141
0.6625 0.7168 0.7724 0.7663 0.7665 0.0807 0.1015 0.0576 0.1105 0.0735 0.0783 0.0703 0.0951 0.0744 0.0658 0.0775 0.1378 0.0938 0.1127 0.1226
0.6667 0.6854 0.7791 0.7800 0.7795 0.7990 0.0850 0.0597 0.1037 0.0543 0.0515 0.0714 0.0566 0.1021 0.0906 0.0791 0.1603 0.0997 0.1140 0.1108
0.6193 0.6901 0.7954 0.7871 0.7436 0.7666 0.8404 0.0960 0.1232 0.0752 0.0660 0.0898 0.0460 0.0707 0.1058 0.0696 0.1410 0.1144 0.0948 0.1175
0.5454 0.6454 0.7321 0.7166 0.7331 0.7609 0.8361 0.7918 0.1004 0.0805 0.0540 0.0902 0.0451 0.1068 0.0924 0.0720 0.1259 0.1018 0.0988 0.1001
0.5594 0.6857 0.7414 0.6629 0.7248 0.6523 0.7804 0.7862 0.7973 0.0906 0.0790 0.0921 0.0481 0.1199 0.0959 0.0650 0.1147 0.0790 0.1291 0.1143
0.5880 0.6145 0.7121 0.7267 0.6943 0.6630 0.7527 0.7069 0.7303 0.7930 0.0978 0.0607 0.0488 0.1357 0.0969 0.0989 0.1437 0.0632 0.1132 0.1148
0.5455 0.6000 0.7029 0.6885 0.6554 0.6926 0.6618 0.7296 0.7446 0.7713 0.8005 0.0739 0.0583 0.1253 0.1082 0.0656 0.1396 0.1021 0.1393 0.1066
0.5770 0.6049 0.6553 0.5227 0.6169 0.5683 0.6525 0.5805 0.6772 0.6869 0.7180 0.7147 0.0421 0.1338 0.0964 0.0602 0.1148 0.1037 0.1060 0.0878
0.5668 0.6023 0.6847 0.6257 0.5574 0.6179 0.6576 0.5748 0.5917 0.7406 0.7422 0.7544 0.8085 0.1488 0.1065 0.0981 0.0996 0.0986 0.0949 0.0722
0.5325 0.5731 0.6141 0.4784 0.5395 0.5387 0.6384 0.5310 0.6705 0.6309 0.6515 0.7252 0.7926 0.7110 0.1178 0.0927 0.0911 0.1137 0.0887 0.0977
0.5039 0.5394 0.6175 0.5436 0.5573 0.5783 0.5886 0.4870 0.6058 0.6844 0.6531 0.7011 0.7850 0.6647 0.8235 0.0619 0.0990 0.0922 0.0998 0.1172
0.5799 0.4831 0.5988 0.4822 0.5807 0.4963 0.6100 0.4051 0.6081 0.6858 0.5913 0.7020 0.7989 0.6506 0.7815 0.8030 0.0902 0.0877 0.0912 0.1250
0.5399 0.4113 0.5397 0.4447 0.5240 0.5427 0.5627 0.3753 0.5512 0.6142 0.4868 0.7001 0.7361 0.6206 0.6931 0.7176 0.7411 0.1161 0.1373 0.1347
0.5427 0.5133 0.5789 0.5249 0.6101 0.5422 0.5875 0.4113 0.6144 0.5876 0.5636 0.6325 0.7244 0.6313 0.6796 0.7146 0.7886 0.7659 0.1224 0.1180
0.5085 0.5428 0.6279 0.5754 0.5676 0.5175 0.6138 0.4533 0.6017 0.5632 0.6322 0.5923 0.6966 0.6234 0.6764 0.6761 0.7688 0.7045 0.7836 0.1122
0.4839 0.5742 0.5863 0.5711 0.5760 0.5578 0.5912 0.4302 0.6911 0.5543 0.5981 0.6211 0.6812 0.6415 0.6357 0.5967 0.7709 0.7132 0.7234 0.7715

Final Accuracy: 0.6185
Backward: -0.1653
Forward:  0.0054

### B.1.5   Model *GEM*

0.1330 0.1199 0.1070 0.0825 0.0609 0.0832 0.1385 0.1123 0.0736 0.1190 0.0666 0.0890 0.0885 0.0723 0.1083 0.0524 0.0976 0.0871 0.1143 0.0743
|
0.7289 0.1426 0.0776 0.1272 0.0637 0.0582 0.0944 0.0961 0.0691 0.0794 0.0822 0.1426 0.0953 0.0792 0.0885 0.1094 0.1328 0.0987 0.1507 0.0976
0.8158 0.8356 0.0601 0.1668 0.0708 0.0624 0.1524 0.0836 0.0994 0.0757 0.0802 0.1278 0.1056 0.0634 0.0625 0.0924 0.1193 0.1390 0.1392 0.1031
0.8134 0.8302 0.8407 0.1509 0.0539 0.0987 0.1566 0.0785 0.1134 0.1025 0.0659 0.0808 0.0746 0.0593 0.0883 0.0759 0.1764 0.1191 0.1253 0.1067
0.7732 0.7940 0.8306 0.7605 0.0622 0.0789 0.1374 0.0669 0.1172 0.0735 0.0979 0.0488 0.0877 0.0802 0.0713 0.1161 0.1434 0.1216 0.1295 0.1032
0.8079 0.8283 0.8478 0.8328 0.7798 0.0797 0.0918 0.0598 0.1376 0.0753 0.0704 0.0516 0.0986 0.0573 0.0549 0.0802 0.1815 0.1077 0.1249 0.1279
0.8262 0.8296 0.8296 0.8471 0.8472 0.8093 0.0888 0.0661 0.0942 0.0890 0.0572 0.0487 0.0707 0.0954 0.0957 0.0793 0.1793 0.1086 0.1306 0.1049
0.8109 0.8214 0.8132 0.8374 0.8445 0.8429 0.8297 0.0984 0.0999 0.0971 0.0699 0.0608 0.0662 0.0722 0.0910 0.0814 0.1720 0.1175 0.1218 0.1165
0.8098 0.8185 0.8268 0.8388 0.8275 0.8391 0.8587 0.7566 0.1019 0.1059 0.0751 0.0727 0.0563 0.1015 0.1116 0.0856 0.1505 0.0991 0.1793 0.1144
0.8260 0.8247 0.8447 0.8290 0.8468 0.8362 0.8517 0.8246 0.8219 0.1096 0.0734 0.0662 0.0595 0.1137 0.0923 0.0855 0.1417 0.1073 0.1398 0.1156
0.8202 0.8264 0.8237 0.8271 0.8462 0.8458 0.8462 0.8307 0.8574 0.8142 0.0670 0.0787 0.0691 0.0957 0.0950 0.0975 0.1403 0.0828 0.1587 0.1204
0.8213 0.8137 0.8228 0.8257 0.8352 0.8517 0.8422 0.8296 0.8515 0.8559 0.8041 0.0808 0.0628 0.1036 0.1005 0.0805 0.1351 0.1107 0.1509 0.1026
0.7878 0.8073 0.8088 0.7913 0.8250 0.8290 0.8328 0.8196 0.8388 0.8343 0.8172 0.7556 0.0686 0.1228 0.0976 0.0922 0.1308 0.1180 0.1360 0.0870
0.8114 0.8181 0.8149 0.8128 0.8328 0.8273 0.8407 0.8254 0.8487 0.8551 0.8339 0.8550 0.8224 0.1212 0.1038 0.1207 0.1185 0.0969 0.1461 0.0928
0.7982 0.8191 0.8133 0.8094 0.8350 0.8343 0.8495 0.8186 0.8473 0.8566 0.8279 0.8552 0.8505 0.7850 0.1130 0.1232 0.1264 0.0816 0.1328 0.1051
0.8044 0.8217 0.8170 0.8198 0.8429 0.8240 0.8490 0.8147 0.8498 0.8568 0.8334 0.8551 0.8489 0.8442 0.8291 0.1096 0.1197 0.0890 0.1746 0.1045
0.8150 0.8198 0.8126 0.8250 0.8348 0.8281 0.8476 0.8191 0.8487 0.8551 0.8343 0.8575 0.8520 0.8447 0.8517 0.8155 0.1148 0.0896 0.1269 0.1269
0.8100 0.8164 0.8030 0.8162 0.8355 0.8242 0.8455 0.8149 0.8348 0.8531 0.8311 0.8541 0.8528 0.8411 0.8558 0.8367 0.8253 0.1083 0.1452 0.1035
0.8031 0.8092 0.8047 0.8170 0.8294 0.8103 0.8421 0.8087 0.8384 0.8479 0.8379 0.8399 0.8467 0.8413 0.8478 0.8377 0.8501 0.8287 0.1335 0.1102
0.7999 0.8107 0.8038 0.8033 0.8168 0.8067 0.8433 0.8044 0.8309 0.8397 0.8355 0.8349 0.8419 0.8320 0.8379 0.8363 0.8421 0.8369 0.7689 0.1119
0.8024 0.8063 0.7966 0.8045 0.8199 0.8153 0.8430 0.8009 0.8317 0.8494 0.8304 0.8391 0.8431 0.8341 0.8426 0.8226 0.8461 0.8436 0.8350 0.8141

Final Accuracy: 0.8260
Backward: 0.0247
Forward:  0.0088

## B.2   MNIST rotations

### B.2.1   Model *single*

0.0903 0.0957 0.0843 0.0882 0.0910 0.0847 0.0998 0.0992 0.0906 0.0756 0.0743 0.0781 0.0839 0.0873 0.0778 0.0810 0.0780 0.0791 0.0884 0.0867
|
0.2777 0.2559 0.2387 0.2382 0.2057 0.1597 0.1548 0.1284 0.1106 0.0912 0.0881 0.0889 0.1082 0.1030 0.0950 0.1011 0.1059 0.1130 0.1263 0.1490
0.4166 0.4466 0.3915 0.3657 0.3048 0.2447 0.2062 0.1701 0.1476 0.1358 0.1285 0.1236 0.1473 0.1221 0.1164 0.1212 0.1074 0.1174 0.1394 0.1646
0.4598 0.5531 0.5332 0.4883 0.4369 0.3510 0.2676 0.2098 0.1822 0.1580 0.1467 0.1379 0.1568 0.1279 0.1240 0.1280 0.1167 0.1215 0.1426 0.1642
0.4760 0.5879 0.5823 0.5779 0.5265 0.4513 0.3448 0.2649 0.2321 0.1786 0.1593 0.1441 0.1541 0.1301 0.1241 0.1240 0.1234 0.1334 0.1529 0.1677
0.4619 0.6080 0.6358 0.6383 0.6069 0.5365 0.4179 0.3122 0.2670 0.1996 0.1727 0.1534 0.1486 0.1271 0.1290 0.1296 0.1319 0.1426 0.1701 0.1892
0.4522 0.6096 0.6652 0.6865 0.6793 0.6383 0.5041 0.3951 0.3331 0.2389 0.1984 0.1681 0.1609 0.1388 0.1443 0.1396 0.1465 0.1489 0.1731 0.1916
0.4191 0.5759 0.6415 0.6779 0.6902 0.6793 0.6057 0.5175 0.4500 0.3221 0.2554 0.1986 0.1720 0.1410 0.1420 0.1406 0.1486 0.1547 0.1770 0.2025
0.4160 0.5657 0.6248 0.6618 0.7004 0.7245 0.6977 0.6576 0.6200 0.4765 0.3857 0.2642 0.2134 0.1595 0.1524 0.1504 0.1471 0.1562 0.1726 0.1965
0.3671 0.5139 0.5866 0.6303 0.6699 0.7021 0.7016 0.6990 0.6878 0.5734 0.4855 0.3443 0.2578 0.1729 0.1503 0.1441 0.1446 0.1525 0.1738 0.1936
0.3269 0.4819 0.5596 0.6152 0.6666 0.7057 0.7219 0.7343 0.7272 0.6526 0.5712 0.4225 0.3191 0.2081 0.1758 0.1648 0.1479 0.1543 0.1665 0.1796
0.3098 0.4567 0.5334 0.5945 0.6441 0.6876 0.7189 0.7482 0.7576 0.7061 0.6630 0.5231 0.4143 0.2643 0.2052 0.1907 0.1557 0.1607 0.1692 0.1877
0.2879 0.4181 0.4965 0.5614 0.6175 0.6678 0.7104 0.7449 0.7595 0.7329 0.7159 0.6327 0.5347 0.3718 0.2821 0.2556 0.1917 0.1882 0.1873 0.1977

```
0.2797 0.4064 0.4695 0.5275 0.5915 0.6495 0.6925 0.7310 0.7486 0.7341 0.7398 0.6979 0.6550 0.5034 0.3892 0.3615 0.2490 0.2378 0.2121 0.2079
0.2545 0.3551 0.4139 0.4677 0.5308 0.5871 0.6519 0.6939 0.7186 0.7197 0.7367 0.7369 0.7174 0.6270 0.5141 0.4821 0.3397 0.3041 0.2487 0.2346
0.2613 0.3557 0.3999 0.4496 0.5069 0.5534 0.6194 0.6540 0.6815 0.6885 0.7134 0.7317 0.7440 0.7064 0.6553 0.6275 0.4707 0.4288 0.3217 0.2804
0.2485 0.3330 0.3686 0.4169 0.4739 0.5138 0.5891 0.6245 0.6503 0.6715 0.6947 0.7258 0.7536 0.7440 0.7201 0.7051 0.5773 0.5326 0.3963 0.3444
0.2468 0.3107 0.3275 0.3603 0.4160 0.4402 0.5168 0.5672 0.5917 0.6307 0.6577 0.7025 0.7393 0.7559 0.7551 0.7474 0.6757 0.6328 0.5115 0.4579
0.2243 0.2914 0.3098 0.3437 0.3873 0.4008 0.4748 0.5190 0.5470 0.5950 0.6142 0.6730 0.7173 0.7530 0.7723 0.7738 0.7373 0.7128 0.5946 0.5345
0.2055 0.2771 0.2953 0.3319 0.3715 0.3848 0.4420 0.4860 0.5122 0.5549 0.5759 0.6355 0.6768 0.7149 0.7563 0.7592 0.7544 0.7478 0.6940 0.6390
0.2136 0.2710 0.2837 0.3126 0.3460 0.3508 0.4033 0.4461 0.4702 0.5237 0.5408 0.6021 0.6513 0.6987 0.7481 0.7590 0.7708 0.7772 0.7434 0.7023
```

```
Final Accuracy: 0.5307
Backward: -0.0896
Forward:  0.4254
```

## B.2.2   Model *independent*

```
0.1013 0.0910 0.1049 0.0761 0.0976 0.1071 0.0960 0.0890 0.0815 0.1172 0.0938 0.1231 0.1140 0.0874 0.0995 0.1005 0.1075 0.0919 0.0942 0.1790
|
0.4221 0.0910 0.1049 0.0761 0.0976 0.1071 0.0960 0.0890 0.0815 0.1172 0.0938 0.1231 0.1140 0.0874 0.0995 0.1005 0.1075 0.0919 0.0942 0.1790
0.4221 0.5266 0.1049 0.0761 0.0976 0.1071 0.0960 0.0890 0.0815 0.1172 0.0938 0.1231 0.1140 0.0874 0.0995 0.1005 0.1075 0.0919 0.0942 0.1790
0.4221 0.5266 0.5069 0.0761 0.0976 0.1071 0.0960 0.0890 0.0815 0.1172 0.0938 0.1231 0.1140 0.0874 0.0995 0.1005 0.1075 0.0919 0.0942 0.1790
0.4221 0.5266 0.5069 0.5967 0.0976 0.1071 0.0960 0.0890 0.0815 0.1172 0.0938 0.1231 0.1140 0.0874 0.0995 0.1005 0.1075 0.0919 0.0942 0.1790
0.4221 0.5266 0.5069 0.5967 0.5091 0.1071 0.0960 0.0890 0.0815 0.1172 0.0938 0.1231 0.1140 0.0874 0.0995 0.1005 0.1075 0.0919 0.0942 0.1790
0.4221 0.5266 0.5069 0.5967 0.5091 0.6481 0.0960 0.0890 0.0815 0.1172 0.0938 0.1231 0.1140 0.0874 0.0995 0.1005 0.1075 0.0919 0.0942 0.1790
0.4221 0.5266 0.5069 0.5967 0.5091 0.6481 0.6278 0.0890 0.0815 0.1172 0.0938 0.1231 0.1140 0.0874 0.0995 0.1005 0.1075 0.0919 0.0942 0.1790
0.4221 0.5266 0.5069 0.5967 0.5091 0.6481 0.6278 0.5654 0.0815 0.1172 0.0938 0.1231 0.1140 0.0874 0.0995 0.1005 0.1075 0.0919 0.0942 0.1790
0.4221 0.5266 0.5069 0.5967 0.5091 0.6481 0.6278 0.5654 0.5874 0.1172 0.0938 0.1231 0.1140 0.0874 0.0995 0.1005 0.1075 0.0919 0.0942 0.1790
0.4221 0.5266 0.5069 0.5967 0.5091 0.6481 0.6278 0.5654 0.5874 0.5711 0.0938 0.1231 0.1140 0.0874 0.0995 0.1005 0.1075 0.0919 0.0942 0.1790
0.4221 0.5266 0.5069 0.5967 0.5091 0.6481 0.6278 0.5654 0.5874 0.5711 0.6846 0.1231 0.1140 0.0874 0.0995 0.1005 0.1075 0.0919 0.0942 0.1790
0.4221 0.5266 0.5069 0.5967 0.5091 0.6481 0.6278 0.5654 0.5874 0.5711 0.6846 0.6723 0.1140 0.0874 0.0995 0.1005 0.1075 0.0919 0.0942 0.1790
0.4221 0.5266 0.5069 0.5967 0.5091 0.6481 0.6278 0.5654 0.5874 0.5711 0.6846 0.6723 0.6533 0.0874 0.0995 0.1005 0.1075 0.0919 0.0942 0.1790
0.4221 0.5266 0.5069 0.5967 0.5091 0.6481 0.6278 0.5654 0.5874 0.5711 0.6846 0.6723 0.6533 0.5998 0.0995 0.1005 0.1075 0.0919 0.0942 0.1790
0.4221 0.5266 0.5069 0.5967 0.5091 0.6481 0.6278 0.5654 0.5874 0.5711 0.6846 0.6723 0.6533 0.5998 0.7046 0.1005 0.1075 0.0919 0.0942 0.1790
0.4221 0.5266 0.5069 0.5967 0.5091 0.6481 0.6278 0.5654 0.5874 0.5711 0.6846 0.6723 0.6533 0.5998 0.7046 0.6965 0.1075 0.0919 0.0942 0.1790
0.4221 0.5266 0.5069 0.5967 0.5091 0.6481 0.6278 0.5654 0.5874 0.5711 0.6846 0.6723 0.6533 0.5998 0.7046 0.6965 0.7011 0.0919 0.0942 0.1790
0.4221 0.5266 0.5069 0.5967 0.5091 0.6481 0.6278 0.5654 0.5874 0.5711 0.6846 0.6723 0.6533 0.5998 0.7046 0.6965 0.7011 0.7532 0.0942 0.1790
0.4221 0.5266 0.5069 0.5967 0.5091 0.6481 0.6278 0.5654 0.5874 0.5711 0.6846 0.6723 0.6533 0.5998 0.7046 0.6965 0.7011 0.7532 0.7424 0.1790
0.4221 0.5266 0.5069 0.5967 0.5091 0.6481 0.6278 0.5654 0.5874 0.5711 0.6846 0.6723 0.6533 0.5998 0.7046 0.6965 0.7011 0.7532 0.7424 0.7131
```

```
Final Accuracy: 0.6241
Backward: 0.0000
Forward:  0.0000
```

## B.2.3   Model *multimodal*

```
0.0910 0.0870 0.0943 0.0850 0.1043 0.1102 0.0820 0.0696 0.1339 0.0871 0.0946 0.0901 0.0961 0.0506 0.0991 0.0819 0.1018 0.0846 0.0778 0.1116
|
0.6383 0.1237 0.1273 0.1238 0.1322 0.1184 0.1173 0.1289 0.1152 0.1068 0.1253 0.1499 0.1202 0.1165 0.1327 0.1088 0.1264 0.0729 0.1397 0.1097
0.6893 0.8122 0.1502 0.1124 0.1077 0.1306 0.1124 0.1104 0.1749 0.0445 0.1239 0.1555 0.0726 0.1460 0.1044 0.0768 0.1344 0.0667 0.0936 0.1140
0.7519 0.7990 0.7989 0.1042 0.0798 0.1415 0.0978 0.0843 0.1692 0.0656 0.1042 0.1048 0.0994 0.1430 0.0601 0.0966 0.1270 0.0688 0.0754 0.1310
0.7601 0.7709 0.7155 0.7898 0.0860 0.1465 0.0952 0.0756 0.1799 0.0509 0.1106 0.1279 0.1067 0.1275 0.0607 0.0947 0.1293 0.0746 0.0888 0.1442
0.6484 0.7890 0.7966 0.8229 0.7745 0.1345 0.0966 0.1006 0.1790 0.0381 0.1081 0.1532 0.1041 0.1488 0.0745 0.0627 0.1352 0.0588 0.1351 0.1240
0.7081 0.7838 0.8047 0.8242 0.7983 0.8155 0.1031 0.1075 0.1842 0.0345 0.0954 0.1556 0.1202 0.1501 0.0727 0.0764 0.1280 0.0617 0.1047 0.1041
0.7024 0.7591 0.8092 0.8247 0.7894 0.8185 0.8534 0.0859 0.1825 0.0374 0.0848 0.1266 0.1013 0.1776 0.0483 0.0851 0.1186 0.0583 0.1074 0.0956
0.6513 0.7391 0.7529 0.8224 0.7751 0.7736 0.7943 0.8205 0.1899 0.0575 0.1190 0.1675 0.1221 0.1437 0.1153 0.0963 0.1484 0.0676 0.1175 0.1182
0.6982 0.7214 0.7352 0.8050 0.7633 0.7760 0.7679 0.7914 0.7938 0.0640 0.0981 0.1094 0.1005 0.1551 0.0467 0.1016 0.1342 0.0753 0.0884 0.1025
0.7137 0.7398 0.7571 0.8232 0.7566 0.8061 0.8019 0.8078 0.7705 0.7824 0.1036 0.1169 0.1033 0.1426 0.0397 0.0948 0.1914 0.0688 0.0853 0.1082
0.6852 0.7492 0.7546 0.8247 0.7722 0.8114 0.8113 0.7814 0.7905 0.7918 0.8244 0.1497 0.1274 0.1331 0.0725 0.0907 0.1988 0.0564 0.0846 0.0966
0.6730 0.7220 0.7138 0.7417 0.7212 0.8037 0.7353 0.7522 0.7307 0.8016 0.7666 0.7150 0.1374 0.1520 0.0688 0.0977 0.1953 0.0512 0.0916 0.0855
0.6708 0.7275 0.7316 0.7189 0.7635 0.8170 0.7439 0.7635 0.7686 0.7958 0.7602 0.7145 0.8476 0.1562 0.0531 0.0954 0.1447 0.0565 0.0712 0.0676
0.7254 0.6934 0.6972 0.7631 0.7215 0.7612 0.7491 0.7587 0.7416 0.7863 0.7467 0.6608 0.8258 0.6332 0.0521 0.0971 0.1256 0.0805 0.0856 0.0970
0.7337 0.6945 0.7183 0.7578 0.7289 0.7751 0.7479 0.7891 0.7405 0.8019 0.7546 0.6818 0.8461 0.6459 0.8378 0.1011 0.1223 0.0799 0.0763 0.0945
0.7242 0.6911 0.7217 0.7509 0.7299 0.7866 0.7370 0.7865 0.7550 0.8301 0.7717 0.6701 0.8528 0.6085 0.8414 0.8089 0.1177 0.0786 0.0751 0.0997
0.7510 0.7034 0.7479 0.7737 0.7465 0.7734 0.7507 0.8094 0.7728 0.8020 0.7903 0.6261 0.8367 0.6149 0.8214 0.7959 0.7532 0.0788 0.0762 0.0928
0.7208 0.6948 0.7312 0.7529 0.7133 0.7838 0.7536 0.8028 0.7382 0.7821 0.7730 0.6797 0.8350 0.6902 0.8278 0.7852 0.7852 0.7722 0.0750 0.0758
0.6976 0.7028 0.7227 0.7413 0.6956 0.7953 0.7429 0.8143 0.7494 0.7935 0.7792 0.6846 0.8354 0.6694 0.8254 0.7987 0.7890 0.7792 0.7901 0.0834
0.7065 0.7008 0.7246 0.7463 0.6862 0.7965 0.7323 0.8055 0.7398 0.7872 0.7736 0.6983 0.8370 0.6515 0.8280 0.7981 0.7994 0.7817 0.7832 0.7928
```

```
Final Accuracy: 0.7585
Backward: -0.0243
Forward:  0.0177
```

## B.2.4   Model *EWC*

```
0.0903 0.0957 0.0843 0.0882 0.0910 0.0847 0.0998 0.0992 0.0906 0.0756 0.0743 0.0781 0.0839 0.0873 0.0778 0.0810 0.0780 0.0791 0.0884 0.0867
|
0.5388 0.4506 0.3594 0.3211 0.2657 0.2081 0.1768 0.1425 0.1281 0.1229 0.1250 0.1321 0.1629 0.1452 0.1390 0.1385 0.1452 0.1508 0.1690 0.1831
0.6677 0.7185 0.6303 0.5681 0.4726 0.3893 0.2711 0.2194 0.2009 0.1661 0.1614 0.1541 0.1547 0.1378 0.1429 0.1309 0.1450 0.1541 0.1775 0.1850
0.6407 0.7390 0.7261 0.6820 0.6034 0.5138 0.3681 0.2737 0.2352 0.1812 0.1661 0.1429 0.1408 0.1319 0.1411 0.1309 0.1466 0.1582 0.1779 0.1820
0.6090 0.7484 0.7841 0.7763 0.7212 0.6385 0.4619 0.3382 0.2771 0.2069 0.1822 0.1554 0.1438 0.1369 0.1459 0.1389 0.1507 0.1570 0.1688 0.1775
0.5821 0.7637 0.8103 0.8164 0.7878 0.7165 0.5352 0.3950 0.3269 0.2367 0.2023 0.1794 0.1677 0.1517 0.1552 0.1510 0.1532 0.1621 0.1781 0.1934
0.4946 0.6821 0.7669 0.8024 0.8114 0.7989 0.6709 0.5325 0.4530 0.3242 0.2628 0.2027 0.1822 0.1570 0.1621 0.1524 0.1534 0.1571 0.1695 0.1820
0.4532 0.6274 0.7089 0.7662 0.7960 0.8216 0.7669 0.6809 0.6191 0.4623 0.3647 0.2488 0.2023 0.1575 0.1548 0.1517 0.1474 0.1533 0.1665 0.1791
0.4296 0.6019 0.6951 0.7478 0.7900 0.8227 0.8156 0.7810 0.7416 0.6254 0.5120 0.3423 0.2714 0.1869 0.1704 0.1602 0.1467 0.1483 0.1484 0.1661
0.3748 0.5299 0.6121 0.6724 0.7289 0.7833 0.8095 0.8095 0.7960 0.7063 0.6105 0.4368 0.3335 0.2148 0.1842 0.1712 0.1496 0.1505 0.1548 0.1645
0.3254 0.4824 0.5717 0.6362 0.7059 0.7562 0.7980 0.8112 0.8081 0.7510 0.6698 0.5132 0.3878 0.2416 0.1922 0.1819 0.1400 0.1382 0.1339 0.1250
0.3239 0.4702 0.5456 0.6064 0.6671 0.7282 0.7805 0.8154 0.8253 0.8043 0.7706 0.6440 0.5137 0.3283 0.2432 0.2252 0.1649 0.1624 0.1561 0.1542
0.2904 0.4132 0.4724 0.5268 0.5831 0.6426 0.7224 0.7650 0.7902 0.7957 0.8104 0.7633 0.6818 0.5040 0.3822 0.3468 0.2227 0.2033 0.1803 0.1648
0.2627 0.3784 0.4305 0.4844 0.5428 0.5957 0.6715 0.7177 0.7479 0.7716 0.8070 0.8100 0.7724 0.6127 0.4755 0.4429 0.2755 0.2451 0.1860 0.1691
```

```
0.2650 0.3579 0.3955 0.4304 0.4782 0.5086 0.5763 0.6263 0.6576 0.6866 0.7372 0.7834 0.8041 0.7512 0.6558 0.6225 0.4341 0.3840 0.2818 0.2343
0.2537 0.3630 0.3971 0.4268 0.4656 0.4821 0.5349 0.5703 0.6037 0.6249 0.6776 0.7357 0.7648 0.7700 0.7470 0.7243 0.5666 0.5086 0.3507 0.2739
0.2460 0.3470 0.3780 0.4074 0.4503 0.4661 0.5114 0.5514 0.5837 0.6105 0.6547 0.7185 0.7585 0.7834 0.7835 0.7782 0.6416 0.5873 0.4241 0.3461
0.2838 0.3438 0.3453 0.3670 0.4010 0.4097 0.4292 0.4794 0.5080 0.5416 0.5934 0.6701 0.7147 0.7673 0.7947 0.7951 0.7438 0.7013 0.5586 0.4891
0.2606 0.3373 0.3489 0.3761 0.4059 0.4112 0.4184 0.4572 0.4866 0.5167 0.5591 0.6303 0.6715 0.7450 0.7909 0.7981 0.7877 0.7728 0.6445 0.5752
0.2598 0.3731 0.4139 0.4396 0.4562 0.4614 0.4630 0.4731 0.4872 0.4995 0.5168 0.5576 0.5771 0.6386 0.6787 0.6911 0.7179 0.7388 0.7337 0.6951
0.2645 0.3793 0.4190 0.4424 0.4693 0.4561 0.4453 0.4455 0.4568 0.4852 0.4992 0.5384 0.5713 0.6441 0.6865 0.6998 0.7426 0.7718 0.7628 0.7414

Final Accuracy: 0.5461
Backward: -0.2047
Forward:  0.5524
```

## B.2.5   Model *GEM*

```
0.0903 0.0957 0.0843 0.0882 0.0910 0.0847 0.0998 0.0992 0.0906 0.0756 0.0743 0.0781 0.0839 0.0873 0.0778 0.0810 0.0780 0.0791 0.0884 0.0867
|
0.7150 0.6438 0.5319 0.4525 0.3707 0.2810 0.1841 0.1309 0.1217 0.1064 0.1065 0.0997 0.0981 0.1055 0.1258 0.1306 0.1641 0.1670 0.1745 0.1746
0.8393 0.8542 0.7718 0.6760 0.5440 0.4304 0.2996 0.2332 0.2152 0.1786 0.1528 0.1297 0.1214 0.1141 0.1268 0.1263 0.1457 0.1592 0.1762 0.1835
0.7994 0.8744 0.8766 0.8386 0.7664 0.6496 0.4421 0.3067 0.2596 0.1915 0.1770 0.1524 0.1490 0.1369 0.1377 0.1357 0.1352 0.1463 0.1414 0.1411
0.8051 0.8883 0.9008 0.8941 0.8503 0.7569 0.5371 0.3685 0.2996 0.2025 0.1667 0.1246 0.1204 0.1062 0.1078 0.1074 0.1174 0.1330 0.1446 0.1531
0.7618 0.8691 0.8882 0.8906 0.8648 0.7938 0.6290 0.4748 0.4036 0.2955 0.2262 0.1661 0.1494 0.1258 0.1334 0.1309 0.1464 0.1583 0.1852 0.1990
0.7982 0.8797 0.9024 0.9039 0.8983 0.8786 0.7858 0.6460 0.5636 0.3800 0.2922 0.1947 0.1705 0.1360 0.1335 0.1306 0.1327 0.1476 0.1699 0.1763
0.7691 0.8492 0.8718 0.8825 0.8834 0.8860 0.8516 0.7770 0.7165 0.5728 0.4280 0.2638 0.2115 0.1425 0.1273 0.1196 0.1041 0.1144 0.1240 0.1389
0.7970 0.8704 0.8938 0.9050 0.9076 0.9135 0.9090 0.8788 0.8498 0.7472 0.6185 0.3839 0.2985 0.1820 0.1537 0.1479 0.1410 0.1439 0.1556 0.1626
0.7733 0.8433 0.8623 0.8763 0.8785 0.8909 0.8900 0.8895 0.8762 0.8190 0.7209 0.5179 0.4132 0.2511 0.1965 0.1865 0.1594 0.1626 0.1621 0.1705
0.8019 0.8729 0.8920 0.9083 0.9062 0.9059 0.9074 0.9033 0.9012 0.8710 0.7910 0.6302 0.4971 0.3003 0.2310 0.2054 0.1636 0.1657 0.1760 0.1859
0.7525 0.8296 0.8507 0.8591 0.8494 0.8492 0.8479 0.8473 0.8460 0.8233 0.8029 0.6590 0.5487 0.3576 0.2717 0.2369 0.1646 0.1594 0.1586 0.1659
0.7470 0.8471 0.8726 0.8832 0.8893 0.8901 0.8932 0.9009 0.9029 0.9076 0.9006 0.8677 0.8089 0.6113 0.4586 0.4161 0.2529 0.2181 0.1831 0.1994
0.7846 0.8612 0.8801 0.8879 0.8874 0.8910 0.8886 0.8959 0.8972 0.8971 0.9064 0.8994 0.8789 0.7509 0.6201 0.5783 0.3766 0.3191 0.2496 0.2356
0.7565 0.8396 0.8660 0.8766 0.8768 0.8783 0.8732 0.8657 0.8646 0.8535 0.8506 0.8571 0.8651 0.8403 0.7835 0.7558 0.5672 0.4996 0.3637 0.3021
0.6846 0.7966 0.8435 0.8663 0.8794 0.8858 0.8868 0.8891 0.8900 0.8844 0.8854 0.8901 0.8952 0.8884 0.8486 0.8184 0.6492 0.5637 0.3904 0.3136
0.7074 0.8058 0.8376 0.8509 0.8513 0.8507 0.8555 0.8575 0.8599 0.8513 0.8539 0.8665 0.8735 0.8757 0.8726 0.8651 0.7586 0.6984 0.5102 0.4074
0.7536 0.8433 0.8657 0.8772 0.8779 0.8779 0.8804 0.8891 0.8928 0.8850 0.8872 0.8869 0.8929 0.8980 0.8933 0.8921 0.8508 0.8196 0.6974 0.5975
0.7565 0.8486 0.8726 0.8781 0.8760 0.8669 0.8700 0.8702 0.8687 0.8578 0.8578 0.8644 0.8730 0.8902 0.8983 0.8995 0.8872 0.8728 0.7845 0.6955
0.7259 0.8321 0.8560 0.8643 0.8649 0.8498 0.8539 0.8634 0.8595 0.8484 0.8412 0.8532 0.8579 0.8719 0.8892 0.8956 0.8968 0.8966 0.8622 0.8130
0.7295 0.8222 0.8484 0.8594 0.8603 0.8568 0.8586 0.8645 0.8654 0.8626 0.8650 0.8621 0.8703 0.8697 0.8817 0.8843 0.8948 0.9046 0.8860 0.8669

Final Accuracy: 0.8607
Backward: 0.0048
Forward:  0.6647
```

## B.3   CIFAR-100 incremental

### B.3.1   Model *single*

```
0.2000 0.2000 0.2000 0.2000 0.1980 0.2000 0.1980 0.1980 0.2000 0.2000 0.2000 0.2000 0.2000 0.2000 0.2000 0.2000 0.2000 0.1980 0.2000 0.2000
|
0.3080 0.1760 0.1580 0.2240 0.1940 0.1960 0.2620 0.2160 0.2020 0.2280 0.1920 0.2280 0.1960 0.1880 0.1680 0.1320 0.2100 0.1940 0.1580 0.2620
0.2860 0.2340 0.1900 0.2260 0.1540 0.2040 0.1620 0.1880 0.2000 0.1960 0.2800 0.1880 0.2360 0.1440 0.1320 0.1260 0.1640 0.1280 0.0880 0.2020
0.3800 0.2440 0.4000 0.2040 0.1840 0.2060 0.1920 0.1100 0.1980 0.2420 0.1900 0.2060 0.2560 0.1560 0.0980 0.1180 0.2040 0.1840 0.1240 0.2600
0.2600 0.1980 0.3480 0.4620 0.2000 0.1920 0.1560 0.1540 0.1960 0.2240 0.2000 0.2420 0.3720 0.1420 0.1480 0.1180 0.1360 0.1900 0.1800 0.1920
0.2060 0.2540 0.3780 0.4220 0.6520 0.2120 0.1240 0.1600 0.2020 0.1780 0.2320 0.1940 0.2400 0.1680 0.1520 0.0820 0.1620 0.1580 0.2160 0.2140
0.2880 0.2500 0.3200 0.3500 0.6480 0.4200 0.1020 0.2020 0.1860 0.1800 0.1480 0.1980 0.2780 0.1180 0.1060 0.1500 0.1860 0.1700 0.2040 0.2200
0.2980 0.2600 0.3260 0.3260 0.4620 0.3260 0.6220 0.1820 0.2100 0.1460 0.1880 0.2740 0.2560 0.1100 0.1140 0.2200 0.2020 0.1880 0.1920 0.0960
0.2680 0.2040 0.3580 0.3820 0.4140 0.3760 0.6020 0.4920 0.2020 0.2120 0.1820 0.2020 0.2860 0.1260 0.0920 0.1480 0.2180 0.1540 0.2100 0.2300
0.3120 0.1980 0.3140 0.3780 0.4300 0.3920 0.5320 0.4500 0.5580 0.1780 0.2920 0.1860 0.2680 0.1280 0.1460 0.2200 0.1880 0.1480 0.1480 0.1940
0.2360 0.2020 0.3940 0.3300 0.4100 0.3860 0.5000 0.3660 0.4000 0.6640 0.2260 0.2220 0.2380 0.1540 0.1080 0.1340 0.1780 0.1220 0.2160 0.3520
0.2740 0.2180 0.2360 0.4260 0.4180 0.3180 0.4040 0.4340 0.3580 0.5860 0.8060 0.1920 0.2580 0.1360 0.1860 0.1260 0.1960 0.1460 0.2520 0.2660
0.3460 0.2440 0.3220 0.3280 0.3840 0.3100 0.4800 0.4260 0.4300 0.5980 0.6780 0.5760 0.2740 0.2000 0.1480 0.1380 0.2300 0.1400 0.1460 0.3140
0.2780 0.2240 0.3500 0.4200 0.4320 0.3540 0.5040 0.4040 0.3060 0.5420 0.6240 0.5060 0.6740 0.1720 0.1560 0.1100 0.2340 0.1980 0.2900 0.1960
0.1940 0.2640 0.3000 0.3380 0.4520 0.3040 0.3920 0.3940 0.3840 0.5840 0.6140 0.4160 0.6020 0.6440 0.1300 0.1380 0.2120 0.1940 0.3220 0.1980
0.2180 0.2840 0.2140 0.3180 0.4820 0.2780 0.4420 0.4160 0.3340 0.4860 0.6000 0.4820 0.6300 0.6240 0.7560 0.1900 0.2340 0.1880 0.3040 0.2360
0.2580 0.2720 0.2780 0.3040 0.4620 0.3060 0.4300 0.3880 0.4020 0.4760 0.4760 0.5320 0.5940 0.5040 0.6840 0.6240 0.2380 0.1680 0.3200 0.3620
0.2440 0.2600 0.2620 0.3760 0.3760 0.2600 0.3760 0.3280 0.2660 0.3900 0.5120 0.5280 0.5400 0.4500 0.6520 0.5200 0.6760 0.1700 0.3280 0.2720
0.2060 0.2520 0.3140 0.3660 0.4380 0.2880 0.3760 0.3820 0.3880 0.4300 0.5400 0.5200 0.6420 0.4660 0.6620 0.5420 0.6020 0.6600 0.2520 0.2080
0.2300 0.2780 0.3200 0.3800 0.4520 0.2760 0.3580 0.4080 0.4040 0.4800 0.5700 0.5300 0.6340 0.4860 0.7240 0.5860 0.5700 0.5640 0.7540 0.2900
0.2560 0.2520 0.3020 0.3900 0.4000 0.2740 0.3220 0.3760 0.3300 0.5240 0.5860 0.4680 0.5980 0.4380 0.6980 0.5140 0.6180 0.4840 0.7000 0.7320

Final Accuracy: 0.4631
Backward: -0.1226
Forward:  -0.0006
```

### B.3.2   Model *independent*

```
0.2000 0.2000 0.2000 0.1980 0.2000 0.1980 0.1980 0.2000 0.1980 0.2000 0.2000 0.2000 0.2000 0.2000 0.1980 0.2000 0.1980 0.2000 0.2000 0.2000
|
0.4180 0.2000 0.2000 0.1980 0.2000 0.1980 0.1980 0.2000 0.1980 0.2000 0.2000 0.2000 0.2000 0.2000 0.1980 0.2000 0.1980 0.2000 0.2000 0.2000
0.4180 0.4140 0.2000 0.1980 0.2000 0.1980 0.1980 0.2000 0.1980 0.2000 0.2000 0.2000 0.2000 0.2000 0.1980 0.2000 0.1980 0.2000 0.2000 0.2000
0.4180 0.4140 0.3640 0.1980 0.2000 0.1980 0.1980 0.2000 0.1980 0.2000 0.2000 0.2000 0.2000 0.2000 0.1980 0.2000 0.1980 0.2000 0.2000 0.2000
0.4180 0.4140 0.3640 0.3200 0.2000 0.1980 0.1980 0.2000 0.1980 0.2000 0.2000 0.2000 0.2000 0.2000 0.1980 0.2000 0.1980 0.2000 0.2000 0.2000
0.4180 0.4140 0.3640 0.3200 0.5620 0.1980 0.1980 0.2000 0.1980 0.2000 0.2000 0.2000 0.2000 0.2000 0.1980 0.2000 0.1980 0.2000 0.2000 0.2000
0.4180 0.4140 0.3640 0.3200 0.5620 0.3000 0.1980 0.2000 0.1980 0.2000 0.2000 0.2000 0.2000 0.2000 0.1980 0.2000 0.1980 0.2000 0.2000 0.2000
0.4180 0.4140 0.3640 0.3200 0.5620 0.3000 0.4100 0.2000 0.1980 0.2000 0.2000 0.2000 0.2000 0.2000 0.1980 0.2000 0.1980 0.2000 0.2000 0.2000
0.4180 0.4140 0.3640 0.3200 0.5620 0.3000 0.4100 0.2880 0.1980 0.2000 0.2000 0.2000 0.2000 0.2000 0.1980 0.2000 0.1980 0.2000 0.2000 0.2000
0.4180 0.4140 0.3640 0.3200 0.5620 0.3000 0.4100 0.2880 0.3440 0.2000 0.2000 0.2000 0.2000 0.2000 0.1980 0.2000 0.1980 0.2000 0.2000 0.2000
0.4180 0.4140 0.3640 0.3200 0.5620 0.3000 0.4100 0.2880 0.3440 0.3880 0.2000 0.2000 0.2000 0.2000 0.1980 0.2000 0.1980 0.2000 0.2000 0.2000
0.4180 0.4140 0.3640 0.3200 0.5620 0.3000 0.4100 0.2880 0.3440 0.3880 0.6120 0.2000 0.2000 0.2000 0.1980 0.2000 0.1980 0.2000 0.2000 0.2000
```

```
0.4180 0.4140 0.3640 0.3200 0.5620 0.3000 0.4100 0.2880 0.3440 0.3880 0.6120 0.4800 0.2000 0.2000 0.1980 0.2000 0.1980 0.2000 0.2000 0.2000
0.4180 0.4140 0.3640 0.3200 0.5620 0.3000 0.4100 0.2880 0.3440 0.3880 0.6120 0.4800 0.5060 0.2000 0.1980 0.2000 0.1980 0.2000 0.2000 0.2000
0.4180 0.4140 0.3640 0.3200 0.5620 0.3000 0.4100 0.2880 0.3440 0.3880 0.6120 0.4800 0.5060 0.5180 0.1980 0.2000 0.1980 0.2000 0.2000 0.2000
0.4180 0.4140 0.3640 0.3200 0.5620 0.3000 0.4100 0.2880 0.3440 0.3880 0.6120 0.4800 0.5060 0.5180 0.4740 0.2000 0.1980 0.2000 0.2000 0.2000
0.4180 0.4140 0.3640 0.3200 0.5620 0.3000 0.4100 0.2880 0.3440 0.3880 0.6120 0.4800 0.5060 0.5180 0.4740 0.4520 0.1980 0.2000 0.2000 0.2000
0.4180 0.4140 0.3640 0.3200 0.5620 0.3000 0.4100 0.2880 0.3440 0.3880 0.6120 0.4800 0.5060 0.5180 0.4740 0.4520 0.4160 0.2000 0.2000 0.2000
0.4180 0.4140 0.3640 0.3200 0.5620 0.3000 0.4100 0.2880 0.3440 0.3880 0.6120 0.4800 0.5060 0.5180 0.4740 0.4520 0.4160 0.3620 0.2000 0.2000
0.4180 0.4140 0.3640 0.3200 0.5620 0.3000 0.4100 0.2880 0.3440 0.3880 0.6120 0.4800 0.5060 0.5180 0.4740 0.4520 0.4160 0.3620 0.4340 0.2000
0.4180 0.4140 0.3640 0.3200 0.5620 0.3000 0.4100 0.2880 0.3440 0.3880 0.6120 0.4800 0.5060 0.5180 0.4740 0.4520 0.4160 0.3620 0.4340 0.4080
```

```
Final Accuracy: 0.4235
Backward:  0.0000
Forward:   0.0000
```

### B.3.3  Model *iCARL*

```
0.2000 0.2000 0.2000 0.2000 0.2000 0.2000 0.1980 0.2000 0.2000 0.2000 0.2000 0.2000 0.2000 0.1980 0.1980 0.1980 0.2000 0.2000 0.1980 0.2000
|
0.3560 0.2000 0.2000 0.2000 0.2000 0.2000 0.1980 0.2000 0.2000 0.2000 0.2000 0.2000 0.2000 0.1980 0.1980 0.1980 0.2000 0.2000 0.1980 0.2000
0.3880 0.5060 0.2000 0.2000 0.2000 0.2000 0.1980 0.2000 0.2000 0.2000 0.2000 0.2000 0.2000 0.1980 0.1980 0.1980 0.2000 0.2000 0.1980 0.2000
0.3840 0.4060 0.5040 0.2000 0.2000 0.2000 0.1980 0.2000 0.2000 0.2000 0.2000 0.2000 0.2000 0.1980 0.1980 0.1980 0.2000 0.2000 0.1980 0.2000
0.3500 0.3900 0.5100 0.5740 0.2000 0.2000 0.1980 0.2000 0.2000 0.2000 0.2000 0.2000 0.2000 0.1980 0.1980 0.1980 0.2000 0.2000 0.1980 0.2000
0.4160 0.3760 0.4320 0.4660 0.6320 0.2000 0.1980 0.2000 0.2000 0.2000 0.2000 0.2000 0.2000 0.1980 0.1980 0.1980 0.2000 0.2000 0.1980 0.2000
0.4820 0.4760 0.4340 0.4940 0.6040 0.5320 0.1980 0.2000 0.2000 0.2000 0.2000 0.2000 0.2000 0.1980 0.1980 0.1980 0.2000 0.2000 0.1980 0.2000
0.4620 0.4240 0.4080 0.4900 0.5540 0.3840 0.6380 0.2000 0.2000 0.2000 0.2000 0.2000 0.2000 0.1980 0.1980 0.1980 0.2000 0.2000 0.1980 0.2000
0.4880 0.4360 0.4600 0.4940 0.5480 0.4520 0.5240 0.5600 0.2000 0.2000 0.2000 0.2000 0.2000 0.1980 0.1980 0.1980 0.2000 0.2000 0.1980 0.2000
0.4620 0.4560 0.3820 0.4480 0.5200 0.4000 0.4800 0.4840 0.5520 0.2000 0.2000 0.2000 0.2000 0.1980 0.1980 0.1980 0.2000 0.2000 0.1980 0.2000
0.4900 0.4480 0.4600 0.4400 0.5340 0.3780 0.4760 0.4620 0.4500 0.7540 0.2000 0.2000 0.2000 0.1980 0.1980 0.1980 0.2000 0.2000 0.1980 0.2000
0.4740 0.4520 0.3700 0.4960 0.3680 0.4400 0.5200 0.4480 0.6120 0.5620 0.7620 0.2000 0.2000 0.1980 0.1980 0.1980 0.2000 0.2000 0.1980 0.2000
0.4400 0.4640 0.3940 0.4480 0.4840 0.3900 0.4680 0.4660 0.4320 0.5620 0.6240 0.5940 0.2000 0.1980 0.1980 0.1980 0.2000 0.2000 0.1980 0.2000
0.5080 0.4740 0.4460 0.4880 0.5020 0.4220 0.5080 0.4940 0.4700 0.5560 0.6680 0.5200 0.7340 0.1980 0.1980 0.1980 0.2000 0.2000 0.1980 0.2000
0.4640 0.4540 0.4320 0.4760 0.5660 0.4240 0.4920 0.4860 0.3860 0.5340 0.6560 0.4480 0.6200 0.6060 0.1980 0.1980 0.2000 0.2000 0.1980 0.2000
0.5360 0.4580 0.3940 0.4380 0.5720 0.4120 0.5140 0.4720 0.4040 0.5040 0.5980 0.4660 0.6140 0.5580 0.7300 0.1980 0.2000 0.2000 0.1980 0.2000
0.5040 0.5280 0.4680 0.4640 0.5460 0.4280 0.5300 0.5020 0.4800 0.5960 0.6660 0.5340 0.6020 0.5120 0.6680 0.6960 0.2000 0.2000 0.1980 0.2000
0.5160 0.5080 0.4800 0.4700 0.5620 0.4040 0.4180 0.5060 0.4700 0.5860 0.6860 0.5320 0.6580 0.4660 0.6940 0.5960 0.7140 0.2000 0.1980 0.2000
0.5580 0.4580 0.4400 0.5320 0.5220 0.4460 0.5000 0.5020 0.4880 0.5480 0.6680 0.5000 0.6560 0.4640 0.5660 0.5000 0.6200 0.6900 0.1980 0.2000
0.5600 0.5160 0.4480 0.4940 0.5780 0.4300 0.4520 0.4900 0.4820 0.5440 0.7160 0.4960 0.6620 0.4760 0.6620 0.4920 0.5760 0.5680 0.7200 0.2000
0.5160 0.4820 0.4240 0.5020 0.6060 0.4260 0.5320 0.5380 0.4620 0.6060 0.6680 0.4920 0.6420 0.4460 0.6600 0.5060 0.5600 0.5280 0.5980 0.7300
```

```
Final Accuracy: 0.5462
Backward: -0.0830
Forward:   0.0000
```

### B.3.4  Model *EWC*

```
0.2000 0.2000 0.2000 0.2000 0.1980 0.2000 0.1980 0.1980 0.2000 0.2000 0.2000 0.2000 0.2000 0.2000 0.2000 0.2000 0.2000 0.1980 0.2000 0.2000
|
0.3080 0.1760 0.1580 0.2240 0.1940 0.1960 0.2620 0.2160 0.2020 0.2280 0.1920 0.2280 0.1960 0.1880 0.1680 0.1320 0.2100 0.1940 0.1580 0.2620
0.2960 0.3380 0.2100 0.2320 0.1380 0.2380 0.1580 0.2260 0.1800 0.2080 0.2080 0.2100 0.2320 0.1820 0.2000 0.1760 0.1800 0.1700 0.1380 0.2220
0.3480 0.2700 0.4340 0.2840 0.2120 0.2080 0.2200 0.0800 0.2000 0.2480 0.2560 0.2020 0.3260 0.2040 0.2380 0.1020 0.2040 0.1820 0.1880 0.1220
0.3040 0.2660 0.4140 0.4840 0.1820 0.2200 0.2180 0.1020 0.1380 0.2360 0.0680 0.2100 0.2440 0.2140 0.1840 0.1220 0.1300 0.1260 0.2880 0.0740
0.2660 0.2060 0.3800 0.4160 0.6040 0.2180 0.1220 0.1400 0.2020 0.2080 0.2120 0.2220 0.1860 0.2900 0.2880 0.1500 0.1780 0.1780 0.1920 0.1160
0.2400 0.2320 0.3880 0.4340 0.4440 0.3460 0.1080 0.1080 0.1560 0.1940 0.2020 0.2020 0.1560 0.1860 0.2680 0.1340 0.1940 0.1700 0.2520 0.1960
0.2720 0.2560 0.3160 0.3820 0.4580 0.2800 0.6100 0.1440 0.1960 0.2040 0.0940 0.2000 0.1900 0.2080 0.3280 0.1540 0.2060 0.1800 0.2640 0.2240
0.3080 0.2840 0.4340 0.5140 0.5340 0.3800 0.5480 0.4720 0.2300 0.2300 0.0800 0.2120 0.2100 0.2740 0.3760 0.0720 0.1920 0.1880 0.2480 0.2560
0.3480 0.2820 0.3600 0.4480 0.4420 0.4240 0.4040 0.3720 0.5440 0.1920 0.1160 0.2460 0.1440 0.2800 0.2120 0.0680 0.1500 0.1860 0.2500 0.2220
0.2680 0.2300 0.3500 0.4400 0.3780 0.4100 0.4340 0.3840 0.4200 0.6560 0.2400 0.2020 0.1480 0.2220 0.2720 0.1700 0.1980 0.1820 0.2060 0.3140
0.3060 0.2760 0.3400 0.4300 0.3900 0.4120 0.3860 0.3920 0.3880 0.5700 0.7860 0.1520 0.2040 0.2380 0.1980 0.1400 0.2160 0.1900 0.2120 0.2000
0.3420 0.2920 0.2980 0.3540 0.3380 0.3780 0.4860 0.3860 0.4500 0.5940 0.6600 0.6240 0.1120 0.2100 0.2080 0.1540 0.2100 0.1780 0.1380 0.1820
0.3080 0.2300 0.3700 0.3940 0.4120 0.4020 0.3380 0.3620 0.4120 0.5980 0.5400 0.4920 0.6720 0.2600 0.1880 0.1120 0.2040 0.1840 0.1620 0.1880
0.2240 0.2280 0.3340 0.4540 0.4640 0.4040 0.4240 0.3820 0.3720 0.5520 0.5620 0.4540 0.6140 0.5840 0.1660 0.1160 0.1840 0.2000 0.1860 0.2080
0.2540 0.2140 0.3240 0.3520 0.3520 0.3880 0.4580 0.3620 0.5400 0.5500 0.4500 0.5700 0.5380 0.7300 0.1340 0.1640 0.1960 0.1540 0.2300
0.3040 0.2540 0.2780 0.3340 0.4240 0.2840 0.3720 0.3320 0.4240 0.4580 0.4740 0.4320 0.4480 0.4480 0.5280 0.5480 0.1540 0.2060 0.2220 0.2880
0.3300 0.2240 0.3360 0.4480 0.4440 0.4280 0.3280 0.3620 0.3620 0.4540 0.6120 0.5300 0.5940 0.4440 0.6120 0.5500 0.6940 0.1980 0.1720 0.2140
0.3460 0.2400 0.3120 0.4440 0.3460 0.3460 0.3740 0.3960 0.4140 0.5240 0.5900 0.5040 0.6440 0.4720 0.5780 0.5260 0.5400 0.6720 0.2620 0.2160
0.2980 0.2140 0.4500 0.4640 0.3960 0.4420 0.3680 0.4340 0.4080 0.5260 0.5740 0.4760 0.6540 0.5340 0.6980 0.5220 0.6360 0.5080 0.7560 0.2160
0.4220 0.2160 0.3580 0.4240 0.3120 0.3980 0.4440 0.4260 0.4120 0.5880 0.6500 0.4640 0.6480 0.5180 0.6980 0.5300 0.5320 0.5180 0.7060 0.7040
```

```
Final Accuracy: 0.4984
Backward: -0.0799
Forward:  -0.0077
```

### B.3.5  Model *GEM*

```
0.2000 0.2000 0.2000 0.2000 0.1980 0.2000 0.1980 0.1980 0.2000 0.2000 0.2000 0.2000 0.2000 0.2000 0.2000 0.2000 0.2000 0.1980 0.2000 0.2000
|
0.5680 0.1720 0.2080 0.2620 0.2900 0.2000 0.2100 0.2160 0.1800 0.2000 0.2120 0.2200 0.2480 0.2240 0.2280 0.2600 0.2320 0.1840 0.1840 0.1600
0.5340 0.4540 0.2200 0.2600 0.2700 0.2100 0.2380 0.1680 0.2020 0.1940 0.2440 0.2080 0.2540 0.1940 0.2920 0.1860 0.1560 0.1960 0.1560 0.1540
0.6280 0.4900 0.4860 0.2300 0.2760 0.2040 0.1740 0.1520 0.2620 0.2300 0.2300 0.1900 0.2880 0.1220 0.2460 0.1680 0.1640 0.1760 0.2040 0.1460
0.6320 0.5960 0.5640 0.6020 0.2880 0.2140 0.2300 0.1340 0.2160 0.2200 0.2040 0.2180 0.1960 0.1320 0.2420 0.2340 0.1580 0.1560 0.1960 0.2120
0.5160 0.5720 0.5540 0.6420 0.7520 0.2040 0.1820 0.1500 0.2520 0.1640 0.1580 0.1620 0.2200 0.1460 0.2100 0.1480 0.1360 0.1480 0.1840 0.1800
0.5020 0.5780 0.5600 0.5620 0.8040 0.6140 0.1580 0.2280 0.2860 0.1920 0.2120 0.2180 0.2300 0.1340 0.2000 0.1280 0.0660 0.1380 0.2160 0.1840
0.5500 0.5400 0.5460 0.5700 0.7620 0.5140 0.7340 0.1520 0.2400 0.1560 0.1760 0.1720 0.2660 0.1500 0.2140 0.1680 0.1200 0.1380 0.2560 0.1800
0.5300 0.6140 0.6080 0.6020 0.7840 0.5800 0.6680 0.5780 0.2620 0.1940 0.1640 0.1740 0.2960 0.1260 0.2020 0.1680 0.0720 0.1820 0.2160 0.1600
0.5000 0.6020 0.6040 0.6420 0.7780 0.5940 0.6400 0.6480 0.6400 0.1760 0.1460 0.1860 0.3020 0.1200 0.1780 0.1460 0.1600 0.1420 0.1840 0.1620
0.5640 0.6020 0.5860 0.6320 0.7100 0.5720 0.6900 0.6280 0.6320 0.7740 0.1840 0.1900 0.2640 0.1640 0.1380 0.1720 0.1360 0.1400 0.1960 0.1400
0.5800 0.6260 0.5940 0.6280 0.7360 0.5460 0.6200 0.6100 0.6240 0.7340 0.7720 0.1740 0.3000 0.1480 0.2060 0.1420 0.1020 0.1620 0.1520 0.1520
0.5480 0.6340 0.6200 0.6040 0.7460 0.5340 0.6840 0.6160 0.6460 0.7520 0.7640 0.6880 0.2540 0.1600 0.1860 0.1480 0.0960 0.1380 0.1800 0.1580
```

```
0.5620 0.6020 0.6480 0.6280 0.7300 0.5940 0.6760 0.6500 0.6340 0.7460 0.7520 0.7100 0.7480 0.1720 0.1800 0.1640 0.0900 0.1700 0.1620 0.1700
0.5440 0.5760 0.6020 0.6060 0.7120 0.5480 0.6720 0.6380 0.6160 0.7240 0.7220 0.6560 0.7780 0.6940 0.1820 0.1740 0.1060 0.1440 0.1960 0.2200
0.5660 0.6140 0.6040 0.6280 0.7340 0.6180 0.6940 0.6100 0.6380 0.7080 0.7200 0.6580 0.7700 0.7140 0.7740 0.1900 0.1120 0.1540 0.1640 0.2060
0.5120 0.5760 0.5940 0.6360 0.7280 0.5660 0.7220 0.6340 0.5840 0.6960 0.6920 0.6120 0.7720 0.6920 0.7580 0.6340 0.1140 0.1240 0.1840 0.1840
0.5840 0.6140 0.6520 0.6340 0.7580 0.5820 0.6740 0.6660 0.6360 0.7340 0.7160 0.6720 0.7460 0.7000 0.7360 0.6820 0.7300 0.1360 0.1880 0.1780
0.5800 0.6100 0.6440 0.6500 0.6800 0.5820 0.7020 0.6440 0.5780 0.7200 0.7340 0.6700 0.7200 0.6760 0.7300 0.7040 0.7620 0.6980 0.1780 0.1860
0.6020 0.6180 0.6640 0.6500 0.7440 0.5980 0.6660 0.6100 0.5900 0.7160 0.7360 0.6580 0.7700 0.6900 0.7400 0.6960 0.7240 0.6720 0.8200 0.1900
0.5640 0.6300 0.6620 0.6780 0.7160 0.5720 0.6740 0.5920 0.6140 0.7320 0.7200 0.6720 0.7640 0.6760 0.7320 0.6860 0.7180 0.6340 0.8080 0.7220

Final Accuracy: 0.6783
Backward: 0.0042
Forward:  -0.0078
```