[Reviews · NeurIPS 2017]

Reviewer 1



The authors of the manuscript consider the continuum learning setting, where the learner observes a stream of data points from training, which are ordered according to the tasks they belong to, i.e. the learner encounters any data from the next task only after it has observed all the training data for the current one. The authors propose a set of three metrics for evaluation performance of learning algorithms in this setting, which reflect their ability to transfer information to new tasks and not forget information about the earlier tasks. Could the authors, please, comment on the difference between continuum and lifelong learning (the corresponding sentence in line 254 seems incomplete)? The authors also propose a learning method, termed Gradient of Episodic Memory (GEM). The idea of the method is to keep a set of examples from every observed task and make sure that at each update stage the loss on the observed tasks does not increase. It seems like there are some typos in eq. (6), because in its current form I don’t see how it captures this idea. GEM is evaluated on a set of real-world datasets against a set of state-of-the-art baselines. I have a few questions regarding this evaluation: 1. comparison to icarl: if I understand correctly, icarl solves a multi-class classification problem with new classes being added over time, i.e. by the end of training on CIFAR-100 it would be able to solve a full 100-classes problem, while GEM would solve only any of 20 5-classes problems. How was the evaluation of icarl performed in this experiment? Was it ever given a hint which of 5 classes to look at? 2. it also seems from table 1 that the results in figure 1 for GEM and icarl were obtained for different memory sizes. If this is true, could the authors comment on why? 3. on MNIST rotations task multi-task method has less negative transfer than GEM, but its accuracy is lower. Could the authors, please, comment on why? Or these differences are non-significant? 4. It would be very interesting to see performance of baseline (multi-task for MNIST and/or multi-class for CIFAR) that gets all the data in a batch, to evaluate the gap that remains there between continuum learning methods, like GEM, and standard batch learning. 5. were the architectures, used for baseline methods (multi-task and icarl), the same as for GEM?

Reviewer 2



The paper proposes an interesting approach to minimizing the degree of catastrophic forgetting with respect to old tasks while retaining a memory buffer of old examples for each task. The experiments are on public datasets relevant to recent work and compare against some reasonable baselines. However, I think the way the paper presents itself is misleading to the readers about where the core innovation of the paper is. I like the way the paper revisits various concepts in section 2, but it largely reads as if the authors are the first to propose these ideas that are already well understood by the community of researchers studying these problems. For example, I do not believe the development of a new term “continuum learning” used constantly throughout the paper is warranted given the fact that it is a very straightforward instantiation of lifelong learning. The concept behind this term continuum learning is not a novel idea. The authors stress the fact that they use a task descriptor, stating it is novel on line 67. I definitely do not think this is novel. Even multi-task learning as explored in the cited (Caruana, 1998) must send a task identifier with the training data in order to choose which task specific layers to use. This component of the system is just not always specifically called out.

Reviewer 3



This algorithm presents a framework for learning in the context of a series of tasks that are presented in a certain order without repeating samples. It offers an approach for dealing with non-iid input data, catastrophic forgetting, and transfer learning as they are required in such a context. After detailing the problem, the authors propose a solution that allows for learning while protecting against catastrophic forgetting and show a simple algorithm for learning based on this framework. They then validate the framework against other recently proposed algorithms as well as single-task and multi-task approaches as backups, and show that their model is less prone to catastrophic forgetting. While I am not familiar with the corpus of work on this particular problem, assuming they are the first to tackle it in this particular context I find their idea to be quite interesting and worthy of inclusion in NIPS this year. The applications of an approach that learns like a human, without needing to review each data point many times and which does not need the tasks to be given simultaneously, are at least clear to me in the case of EEG processing and surely have other interesting applications as well. My one concern has to do with the validation datasets. As they do not use the standard MNIST and CIFAR100, I find it important to see what a benchmark accuracy might be (trained without the rather strong constraint of only seeing each data point once) to get a sense of where this method currently is in terms of performance. Further, there are some issues with the paper that I would like to better understand: 1. How long to the various methods take? As is pointed out, the constraints of their gradient steps require an inner optimization and it would be interesting to see how that compares time-wise with more standard approaches. Figure 1 has 'training time' on the x-axis, but is this in terms of the number of samples seen or the actual clock time? 2. Because they make a claim based on the relative size of p and M, it would be nice to know exactly how big your CIFAR100 network is.